# Defect engineered bioactive transition metals dichalcogenides quantum dots

Xianguang Ding[1,2], Fei Peng[1], Jun Zhou[3], Wenbin Gong[4], Garaj Slaven [2,3], Kian Ping Loh [2,5], Chwee Teck Lim [2,6,7,8] & David Tai Leong [1]

Transition metal dichalcogenide (TMD) quantum dots (QDs) are fundamentally interesting because of the stronger quantum size effect with decreased lateral dimensions relative to their larger 2D nanosheet counterparts. However, the preparation of a wide range of TMD QDs is still a continual challenge. Here we demonstrate a bottom-up strategy utilizing TM oxides or chlorides and chalcogen precursors to synthesize a small library of TMD QDs ($MoS_2$, $WS_2$, $RuS_2$, $MoTe_2$, $MoSe_2$, $WSe_2$ and $RuSe_2$). The reaction reaches equilibrium almost instantaneously (~10–20 s) with mild aqueous and room temperature conditions. Tunable defect engineering can be achieved within the same reactions by deviating the precursors' reaction stoichiometries from their fixed molecular stoichiometries. Using $MoS_2$ QDs for proof-of-concept biomedical applications, we show that increasing sulfur defects enhanced oxidative stress generation, through the photodynamic effect, in cancer cells. This facile strategy will motivate future design of TMDs nanomaterials utilizing defect engineering for biomedical applications.

[1] Department of Chemical and Biomolecular Engineering, National University of Singapore, Singapore 117585, Singapore. [2] Centre for Advanced 2D Materials, Graphene Research Centre, National University of Singapore, Singapore 117546, Singapore. [3] Department of Physics, National University of Singapore, Singapore 117542, Singapore. [4] Division of Advanced Nanomaterials, Suzhou Institute of Nano-Tech and Nano-Bionics, Chinese Academy of Sciences, Suzhou 215123, China. [5] Department of Chemistry, National University of Singapore, Singapore 117543, Singapore. [6] Department of Biomedical Engineering, National University of Singapore, Singapore 117575, Singapore. [7] Mechanobiology Institute, Mechanobiology Institute, Singapore 117411, Singapore. [8] Biomedical Institute for Global Health Research and Technology, Singapore 117599, Singapore. Correspondence and requests for materials should be addressed to C.T.L. (email: ctlim@nus.edu.sg) or to D.T.L. (email: cheltwd@nus.edu.sg)

Two-dimensional (2D) transition metal dichalcogenides (TMDs) with comparable structures to graphene, have attracted tremendous attention during the past few years. This large family of layered materials shared a common molecular formula and general structure where each member consists of mono-atomic thick stacked layers of repetitive covalently bonded X–M–X (M = Transition metal; X = Chalcogen). With decreasing number of layers, TMDs nanosheets transit from an indirect gap to a direct band-gap semi-conductor (e.g., 1.2–1.9 eV for $MoS_2$ from bulk form to monolayer)[1,2]. This special layers-dependent physical property inspired 2D TMDs nanosheets-enabled applications in biomedicine[3,4], sensors[5], transistors[6], catalysts[7,8], photodetectors[9], and energy storage devices[10,11]. Further reduction of the lateral size of TMDs few-layers or monolayer nanosheets to quantum dots (QDs) or 0D nanodots further accentuate their electrical/optical properties due to a stronger quantum confinement and edge effects[12]. This enhancement introduced more dimensions of interesting and exploitable properties for catalytic and biomedical applications[13,14], of which the latter is gravely under-represented. This could be due to the harsh and non-biocompatible methods in robust synthesis of many of the TMD QDs.

Atomic defects in transition metal dichalcogenides (TMDs) nanomaterials add an additional dimension to its optical, chemical and electronic activities. Top-down approaches like laser, plasma and electron bombardment have been intensively used to introduce defects on TMDs nanosheets and thereby controlling their optical and electrical properties. Moreover, the methods are usually single sheet process, albeit with great precision but at the expense of low scalability and high cost. Applying the same top–down approaches to reliably engineer defects in TMD QDs is extremely difficult due to the minute size of the QDs. This lack of defect engineering option has further curtailed TMD QDs' potential.

In the past few years, many strategies, including lithium intercalation and exfoliation have been developed for synthesizing 2D TMD nanosheets[15–17], but mild and general synthesis methods for 0D TMDs QDs are few. Top-down synthesis methods through a combination of grinding and sonication techniques, could obtain 1T-phase TMD QDs[11,12]. Although much progress has been achieved, to date, many more problems remain that needed solving before exploring their unique properties for catalytic and biomedical-based application. The top-down production of high quality TMD QDs has so far been extremely challenging because of the high degree of size and layers reduction required resulting in a low yield of production at the end of a long tedious process[18,19], which involve time-consuming exfoliation and ultrasonication processes starting from bulk TMD crystals, exfoliated first to 2D TMD nanosheets and then further cutting them down to TMD QDs sizes. Fastidious post-treatment processes (for example, gradient centrifugation) is required; further reducing the yields of each sized TMDs QDs. Reports on bottom-up strategies for TMD QDs are generally limited to harsh hydrothermal methods (200 ºC and at least 24 h)[20–22]. Further defect engineering of as-synthesized TMD QDs will therefore allow for more property explorations of TMD QDs for catalytic, semi-conducting and biomedical applications.

Here we demonstrate a facile and possibly universal bottom-up route to synthesize $MoS_2$ QDs with different degrees of defect under a fair degree of control. All the $MoS_2$ QDs synthesized under our mild aqueous condition are of uniform sizes of around 3.9 nm. This synthesis method has been further expanded to successfully prepare a library of various other TMD QDs. We also showed, using $MoS_2$ QDs, its potential as a photodynamic agent that is easily tunable through bottom-up disordering engineering.

Previous studies have alluded to the possibility of defects in semiconductor QDs matter to photodynamic effect but that was never proven[23], possibly due to the difficulty in controllable engineering defects in semiconductor QDs. In this paper, different degrees of defect engineering were achieved in 0D TMDs QDs via bottom-up stoichiometry deviations. With this defect-variable TMD QDs, the suspected correlation between defect degree and photodynamic properties is verified. The possible mechanism of defect enhanced reactive oxygen species (ROS) generation was proposed. By bottom-up defect engineering of TMD QDs, we showed that we could tune photodynamic oxidative stress effect on cancer cells.

## Results

**Characterization of as synthesized $MoS_2$ QDs.** The $MoS_2$ QDs were prepared by a bottom-up route conducted in aqueous condition at room temperature via a simple chemical reaction starting with $Na_2S$ and $MoCl_3$, $MoCl_5$ or $MoO_3$. Inspired by biomineralization, natural biopolymers such as BSA were used as surfactant because of their template effect and excellent biocompatibility[24,25]. Briefly, pH value of Mo-precursor solution was firstly adjusted to be above 11. In this high pH solution, the molybdenum precursors—$MoCl_5$ or $MoO_3$ decompose, yielding stable $MoO_4^{2-}$ (Supplementary Figure 1). The reaction scheme can be expressed as:

$$4MoCl_5 + O_2 + 6H_2O + 8OH^- \rightarrow 4MoO_4^{2-} + 20HCl$$

$$MoO_3 + 2OH^- \rightarrow MoO_4^{2-} + H_2O$$

Then they were transferred into BSA solution; $Na_2S$ solution was introduced into the mixture. Subsequent adjusting of pH value to 6 with addition of HCl activates the sulfur precursors, initiating the reaction between $MoO_4^{2-}$ and sulfur precursors protected by BSA (Fig. 1a, Supplementary Movie 1). Our one-pot reaction was carried out under mild experimental conditions in the laboratory; without the need for high temperature and pressure or special apparatus[26]. Our method is easily scalable. The aqueous product showed a faint yellow color with good dispersibility (Fig. 1b). The transmission electron microscopy (TEM) images showed a consistent size quality of the as synthesized $MoS_2$ QDs to be ~3.9 nm (Fig. 1c). The observed d-spacing (0.27 nm) is assigned to the (100) atomic plane of $MoS_2$, confirming the high crystallinity of the $MoS_2$ QDs (Fig. 1d). X-ray photoelectron spectroscopy (XPS) was also performed to measure the composition and the corresponding chemical valence of the elements in $MoS_2$ QDs (Supplementary Figure 2). Typical $Mo^{4+}$ $3d_{3/2}$ peaks of 232.8 eV and $Mo^{4+}$ $3d_{5/2}$ peaks of 229.6 eV could be clearly observed from the XPS spectrum (Supplementary Figure 2a), which suggested the presence of Mo (IV) of $MoS_2$ QDs[27]. Supplementary Figure 2b shows the binding energy of sulfur with the S $2p_{3/2}$ and S $2p_{1/2}$ peaks at 162.15 eV and 163.05 eV, which is consistent with −2 oxidation state of sulfur. The atomic ratio of Mo to sulfur was quantified to be 1: 2.31. The higher sulfur content compared to the initial feed ratio (Mo: S = 1:2) might be due to the presence of S- containing cysteines in BSA on the surface of $MoS_2$ QDs. The structure of $MoS_2$ QDs was further characterized by powder X-Ray diffractometry (XRD, Fig. 1e). The XRD pattern of $MoS_2$ QDs shows broad diffraction peaks, which is in accordance with the XRD features of low-dimensional nanoparticles, suggesting the small grain sizes of the $MoS_2$ QDs. The main peaks at 11º, 29º and 36º can be assigned to the characteristic (001), (100) and (103) planes of hexagonal 2H-$MoS_2$ (JCPDS NO. 24-0513). The colloidal suspension of $MoS_2$ QDs prepared was very stable, with no obvious aggregations over at least 3 months (Supplementary Figure 3). This high stability property is an important advantage

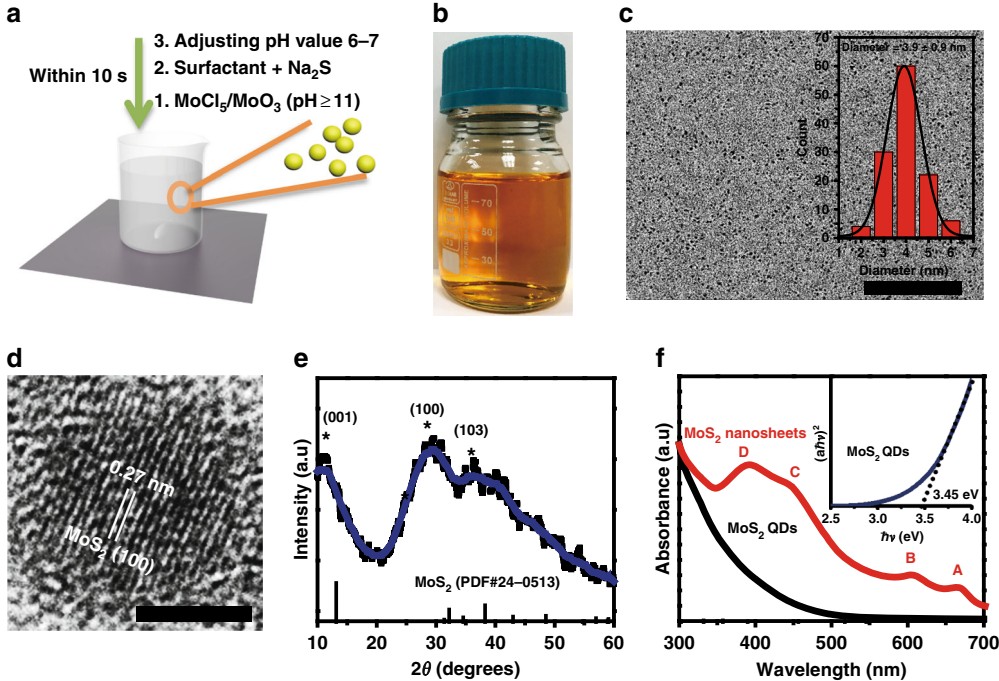

**Fig. 1** Benign aqueous room temperature bottom-up synthesis of MoS$_2$ QDs. **a** Preparation of MoS$_2$ QDs with bottom-up strategy. **b** Easy and scalable aqueous MoS$_2$ colloidal suspension. Synthesis step takes less than 10–20 s (also refer to Supplementary Movie 1). **c** Ultrasmall but consistent as-synthesized MoS$_2$ QDs. Inset: Distribution of MoS$_2$ QDs size measured (ImageJ, $n = 200$). Scale bar: 100 nm. **d** HRTEM (Scale bar: 5 nm), **e** XRD and **f** UV-vis absorption spectra of MoS$_2$ QDs showing a distinctively different nanomaterial from exfoliated MoS$_2$ nanosheets

in biomedical applications. In comparison to 2D MoS$_2$ nanosheets with relatively large lateral dimensions, four distinct excitonic peaks from 2D MoS$_2$ nanosheets (A, B, C and D) were absent in the optical spectrum of MoS$_2$ QDs (Fig. 1f), along with strongly absorption blue-shifted towards shorter wavelengths. These optical features should result from the stronger quantum size effect of MoS$_2$ QDs with decreased lateral dimensions relative to 2D MoS$_2$ nanosheets (Supplementary Figure 4). Noticeably, MoS$_2$ QDs exhibited a new absorption peak at around 310 nm. Similar absorption features were also observed by others[28,29]; attributed to their quantum confinement effect. Here the as-prepared MoS$_2$ QDs had an average diameter of 3.9 nm, which is close to the bulk Bohr exciton diameter of ~4 nm[30]. The discretized bands induced by quantum confinement allowed transitions from the deep valance band to the conduction band, increasing the discrete absorption bands of MoS$_2$ QDs. By using the method based on the relation of $(\alpha\hbar\nu)^2$ versus $\hbar\nu$ ($\alpha$ is absorbance, $\hbar$ is the Planck's constant and $\nu$ is frequency), the optical band gap of our MoS$_2$ QDs was calculated to be ~3.45 eV, which is higher than that of bulk and single-layer MoS$_2$ (1.2–1.9 eV). The quantum confinement effect and surface states (like surface defects) induced by the bottom-up method in MoS$_2$ QDs may have collectively contributed to this band gap. With the presence of localized defect states, the surface states can narrow the calculated optical band gap through generating band-tailing effects[31–33].

**Surfactant effect on quality of MoS$_2$ QDs.** The hydrodynamic diameter of MoS$_2$ QDs was determined to be 13 nm by dynamic light scattering (DLS), which was slightly larger than that of BSA (10 nm) and significantly higher than the TEM size of ~3.9 nm (Fig. 2a). One

MoS$_2$ QD is likely to be interacting with one BSA molecule (Fig. 2b). The estimation value from thermogravimetric analysis was slightly larger than the DLS result (Supplementary Figure 5).

With BSA as template to confine the growth of MoS$_2$ QDs, the biomineralization process allow good size quality control. To investigate the role that the surfactant plays in the quality of MoS$_2$ QDs, we also tested gluconate (Glu), poly-arginine (Poly-Arg) and cysteine (Cys) in the direct synthesis method. These biomolecules were selected as they provide the representative functional groups of BSA respectively: in Glu (–COOH), in Poly-Arg (–NH$_2$), in Cys (–SH, –COOH, –NH$_2$). Figure 2c suggests that all of above surfactants could mediate the successful synthesis of MoS$_2$ QDs, while particle quality is highly dependent on the surfactant of choice. To quantify the particle size distribution, full width at half maximum (FWHM) was counted by analyzing the poly-distribution of MoS$_2$ QDs from different surfactants. The size distribution and poly-distribution were analyzed from TEM samples with over 200 particles (with ImageJ). The statistical FWHM values of MoS$_2$ QDs from BSA, Cys, Glu and Poly-Arg were 1.8, 3.1, 5.8 and 7.2, with the mean sizes of 3.9, 7.4, 7.3 and 9 nm, respectively (Fig. 2d), suggestive that BSA was more favorable for smaller QDs size and size distribution control. This also ruled out the sole fine size control of –COOH, –NH$_2$ and –SH groups alone. Previous research suggested that disulfide bonds might possess much stronger binding energy to 2D MoS$_2$ compared to single thiol[34,35]. To check that possibility in our MoS$_2$ QDs, first-principle calculation was carried out to verify the binding affinity of the major functional groups of BSA to MoS$_2$ QDs. As shown in Fig. 2e, benzene rings and disulfide exhibited much higher binding energy to MoS$_2$ QDs compared to –SH, –COOH, –SH, –NH$_2$ and –OH groups, which is 1.3 eV and 0.84 eV with respect to that of other groups (0.46 eV for –SH, 0.45 eV for –NH$_2$ and 0.51 eV for –COOH) (Supplementary Table 1). These simulation results were consistent with the statistic FWHM values from TEM images (Fig. 2c), indicating that benzene rings and disulfide bonds may provide stronger affinity to stabilize QDs against aggregation and confine them for homogeneous growth, thus encouraging overall seeding and

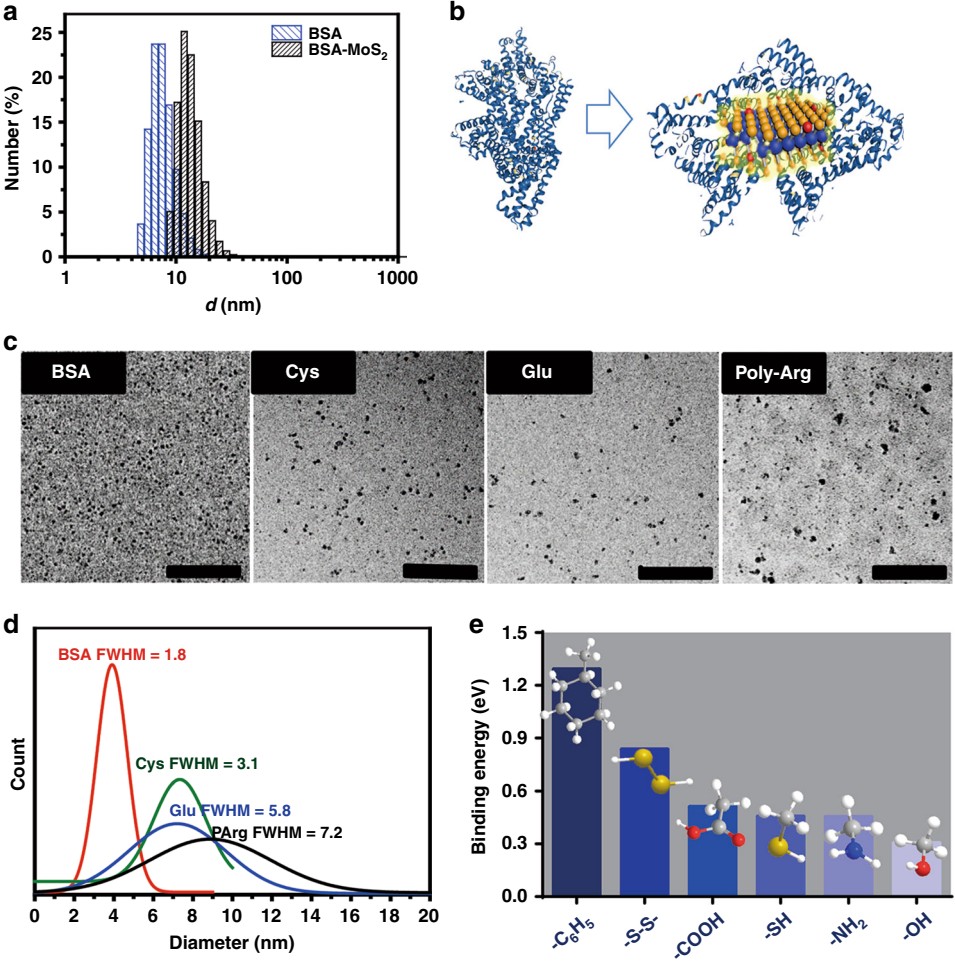

**Fig. 2** Surfactant effect on quality of MoS$_2$ QDs. **a** Size of the BSA and MoS$_2$ QDs determined by DLS. **b** Schematic illustration of possible BSA interactions with one MoS$_2$ QD. **c** TEM images of MoS$_2$ QDs synthesized with BSA, Cys, Glu and Poly-Arg. Scale bar: 200 nm. **d** Statistical analysis of the size distributions of MoS$_2$ QDs synthesized with different biological surfactants. **e** Binding energy of major functional groups of BSA to MoS$_2$ QDs

growth process in our case. To get direct experimental evidence that disulfide bonds does make a significant difference in the synthesis of MoS$_2$ QDs, disulfide bridges of BSA was first reduced with NaBH$_4$ to sulfhydryl groups. These denatured BSA (dBSA) with thiol groups were then used to prepare MoS$_2$ QDs as per our synthesis protocol. The size of the as-acquired MoS$_2$ QDs using dBSA exhibited a significant increase from 3.9 nm to a mean size of 6.3 nm (Supplementary Figure 6). This result was similar to Cys mediated samples which are laden with thiol groups, verifying the significant effect of disulfide bonds on the fine size distribution control of MoS$_2$ QDs. The effect of benzene rings was also tested by using 2, 5 dihydroxybenzoic acid (DBA) as a stabilizer to synthesize MoS$_2$-DBA. However, compared with other surfactants, the morphology of MoS$_2$-DBA QDs was less controllable with the FWHM value to be 13.4 (Supplementary Figure 7), possibly due to the strong van der Waals interactions between benzene rings and weak ions-benzene ring interactions in the seeding stage of the particle formation. From these results we can conclude that the optimal surfactant for synthesizing MoS$_2$ QDs are molecules containing disulfide bonds such as BSA.

The formation of MoS$_2$ QDs was very fast at relatively low energy, as observed by TEM images sampled at different reaction time points and synthesis temperature (Supplementary Figure 8 and 9). The acquired MoS$_2$ QDs could be easily purified by the addition of Cu$^{2+}$ solution to precipitate out the MoS$_2$ QDs,

followed by centrifugation and dialysis to remove the supernatant and ions[36] (Supplementary Figure 10), thus avoiding the fussy and time-consuming post-treatment process.

**Expansion of this synthetic strategy to other TMD QDs**. We expanded this facile and mild bottom-up method to prepare a variety of TMD QDs based on the straightforward chemical reaction between chalcogen precursor and transition metal salt such as metallic oxide or metallic chloride (Fig. 3a), which may be applied to synthesize various TMD QDs (Fig. 3b). To validate the feasibility of this method, BSA was applied as the surfactant for the proof-of-concept verification. Different transition metal sources (for example Mo, W and Ru) and chalcogenide source (S, Se and Te) were mixed with BSA at the pH value of 11. As suggested in Fig. 4, after pH value was adjust to 6 ~ 7. WS$_2$, RuS$_2$, MoTe$_2$, MoSe$_2$, WSe$_2$ and RuSe$_2$ were successfully prepared. The TEM clearly illustrated their highly homogeneous of size distribution with majority of the diameters to be below 10 nm. The corresponding images of different TMD QDs suspensions in H$_2$O showed good dispersibility. Because of different bandgaps and sizes, the prepared TMD QDs library showed different colors in aqueous solution and exhibited correspondingly different UV-Vis absorption spectra (Supplementary Figure 11). It is worth noting that some rod-like shape can also be found in the TEM

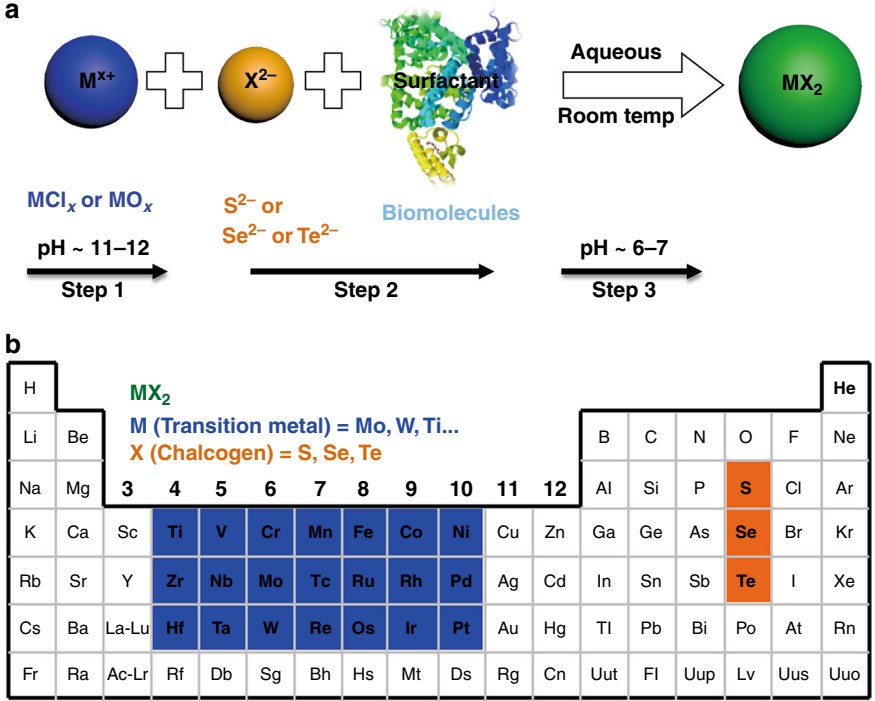

**Fig. 3** Illustration of the bottom-up synthesis of TMD QDs under mild condition. **a** In this study, only the synthesis of commonly reported TMDs ($MoS_2$, $WS_2$, $RuS_2$, $MoTe_2$, $MoSe_2$, $WSe_2$ and $RuSe_2$) QDs were showed. **b** Extrapolating the synthesis method to other TMD QDs synthesis may present an universal method to synthesis the various TMDs QDs

images of $MoTe_2$ samples, possibly due to high reactivity and instability of $MoTe_2$ compounds compared to $MoS_2$ /$MoSe_2$ compounds[37].

**Stoichiometry and characterization**. As our approach is bottom-up, the relative molarity of the reagents can be varied to be different from their stoichiometric ratios, enabling defect creation or dopant incorporation[38]. This theoretically allowed us to engineer defects in the as-synthesized TMD QDs from the bottom-up manner, thus providing a window of opportunity to tune their optical and electrical properties at the synthesis stage. Using $MoS_2$ QDs as an example, three Mo/S relative molar concentrations, (4:2, 4:4 and 4:8) were pre-determined and reacted. Their respective products were named as $MoS_2$-$D_H$, $MoS_2$-$D_M$ and $MoS_2$-$D_L$ respectively. As shown in Fig. 5a, with increasing sulfur amount, the sample color changed from pale yellow for $MoS_2$-$D_H$ to yellow and orange for $MoS_2$-$D_M$ and $MoS_2$-$D_L$, respectively. Careful size analysis from TEM images (~200 particles counted) from the three groups indicated that varying initial relative concentration of reagents did not change the size distributions of the as-synthesized $MoS_2$ QDs. Three $MoS_2$ QDs groups exhibited similar size distributions with the average diameter of around 3.9 nm (Fig. 5b). The HRTEM images showed obvious discontinuities on the lattice planes such as dislocations and distortions were observed in the groups with the off-stoichiometric $MoS_2$ S mole ratios like $MoS_2$-$D_H$ and $MoS_2$-$D_M$ QDs group (Fig. 5b). This discontinuity of the local lattice was thought to be caused by defects within crystals[39]. In $MoS_2$-$D_H$ HRTEM images, several lattice planes with more discontinuities presented in the $MoS_2$ structures, suggesting the strongly disordered

arrangement of nanodomains in $MoS_2$-$D_H$ samples (Fig. 5b). When the sulfur amount increases approaching stoichiometric $MoS_2$, relatively perfect single crystals without disorder were observed, suggesting the relatively fewer defects in $MoS_2$-$D_L$. (Fig. 5b) The XRD diffraction peaks in three samples were broadening, confirming the nanoscale of these crystallites in different dimensions (Fig. 5c). A slight shift of major diffraction peaks indexed to (100) and (103) toward lower angles could also be observed from $MoS_2$-$D_H$ to $MoS_2$-$D_L$ samples, indicating the gradually enlarged lattice constants with increasing sulfur amounts and the homogeneous phase structures of all three $MoS_2$ QDs groups. The stoichiometry related surface states were also investigated by studying the PL behavior of three $MoS_2$ QDs. All three defect states exhibited excitation-dependent PL behaviors (Supplementary Figure 12). Their representative PL spectrum, with each emission spectrum were normalized to their solution optical absorbance density at the PL excitation wavelength of 400 nm (Fig. 5d). Such normalization was used to highlight the PL intensity change. By integrating the respective peak areas, we observed that the quantum yield (QY) of $MoS_2$-$D_H$ (least sulfur content) possess the highest QY, which is approximately 1.79 times more than that of $MoS_2$-$D_M$ (medium sulfur content) and 2.42 times higher than that of $MoS_2$-$D_L$. Previous reports suggested that the surface state of nanomaterial is similar to a molecular state. Both the surface state (such as defects) and intrinsic state (quantum size effect) of nanomaterial contribute to the complexity of excited states of QDs[40,41]. The TEM analysis of the lattice planes disorder appears to point to $MoS_2$-$D_H$ as having the greatest degree of defects (Fig. 5). Interestingly, we observed a trend that with more defects, the photoluminescence quantum yield also increased. Moreover, the $MoS_2$ QDs show two emission peaks with one at around 463–478 nm and the other one located at 530 nm, which could be attributed to intrinsic state emission (electron-hole recombination) and defect state emission, respectively[29]. In general, both intrinsic and defect state emissions influence the fluorescence spectrum. The red shift of fluorescence peaks from 463 nm of $MoS_2$-$D_L$ to 469 nm for $MoS_2$-$D_M$ and 478 nm for $MoS_2$-$D_H$ can be observed and possibly due to the gradually increasing defects from $MoS_2$-$D_L$ to $MoS_2$-$D_H$.

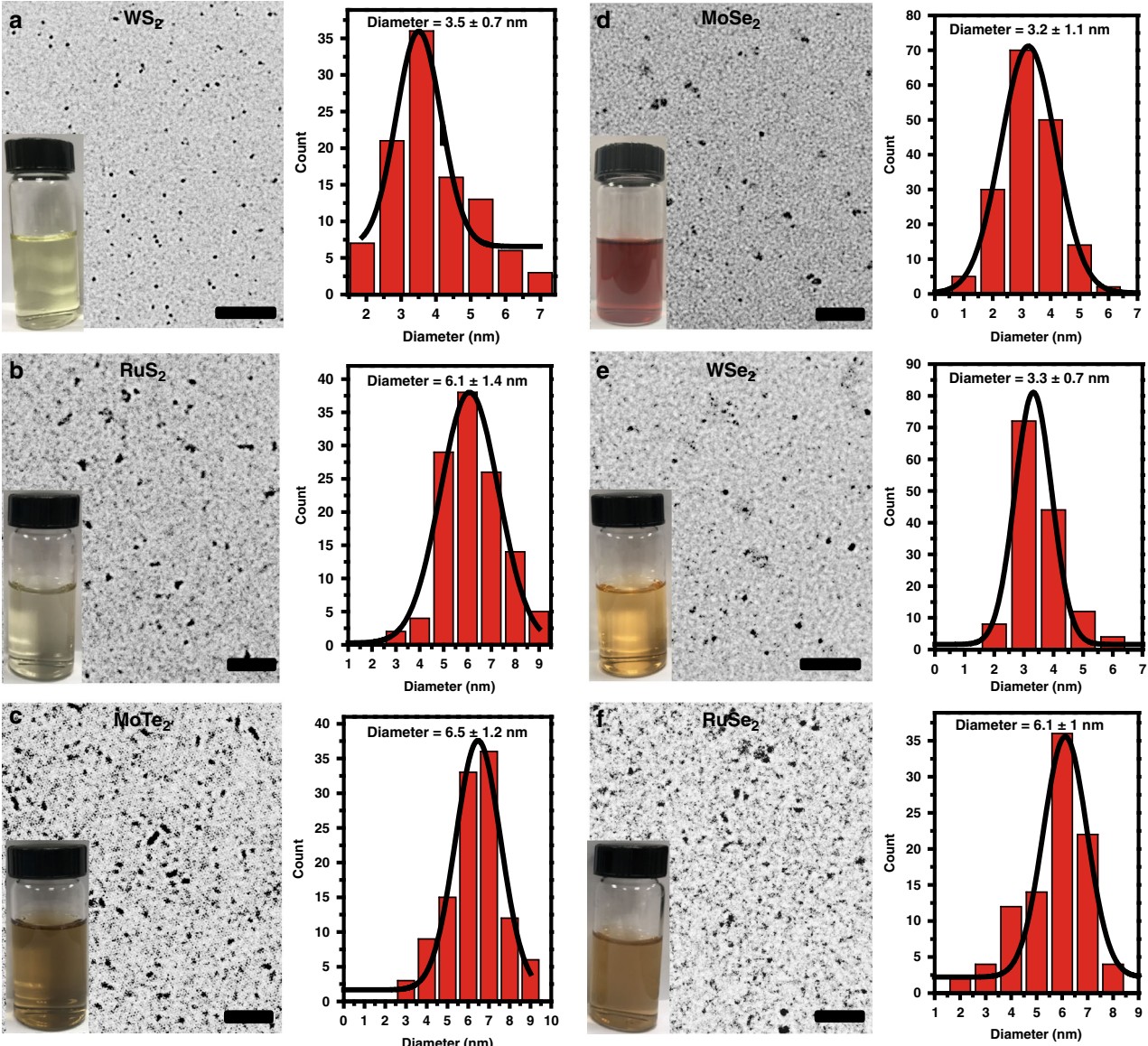

**Fig. 4** TEM images of other TMDs QDs synthesized under similarly mild conditions. **a–f** TEM images of WS$_2$, RuS$_2$, MoTe$_2$, MoSe$_2$, WSe$_2$ and RuSe$_2$ QDs. Insets: Stable colloidal suspension appearance of TMDs dispersed in aqueous condition. TMDs QDs have consistent sizes. Scale bar: 100 nm

The controversy of whether defect sites and photoluminescence quantum yields are positively or negatively correlated is still ongoing. The influences of surface defects on photoluminescence quantum yields in smaller sized MoS$_2$ quantum dots is still largely unknown. From our observations, the intrinsic state emission of the highest to lowest defect sites groups, MoS$_2$-D$_H$ to MoS$_2$-D$_L$ QDs still play the leading role in the PL emission and especially showed significantly higher efficiency in MoS$_2$-D$_H$ QDs. One of the possible explanation is the passivation effect from oxygen atoms in the MoS$_2$ crystal lattice. In the presence of oxygen, the PL of CdSe QDs could be enhanced by as much as a factor of 6, resulting from the surface passivation by oxygen on nanocrystalline surfaces[42]. This phenomenon of oxygen atoms induced surface passivation of QDs has also been identified in many other semiconductor QDs[43,44]. The embedded oxygen atoms in the crystalline of MoS$_2$ structures, presumably played two roles; first by creating sulfur distortion defects which support defect state emission, second by forming Mo–S–O bond on the crystalline surface, which passivate it thus enabling the intrinsic state emission

enhancement. Preliminary experiment on the sulfur defect (in the form of Mo–O) reveal that the defect could increase PL intensity. However, substantial work such as ultrafast dynamics studies can to explore these photophysics of MoS$_2$ QDs in greater detail.

To obtain more information about the intrinsic changes of surface states, high-resolution XPS can reveal the surface chemical environment of the MoS$_2$ samples. Figure 5e shows two characteristic peaks located at ~ 229.6 and ~232.8 eV in three spectrums, which arose from the Mo$^{4+}$ 3d$_{5/2}$ and Mo$^{4+}$ 3d$_{3/2}$, suggesting the dominance of Mo (IV) in the MoS$_2$ samples. Compared with MoS$_2$-D$_L$, samples with less sulfur amount show significantly broadened peaks from MoS$_2$-D$_M$ to MoS$_2$-D$_H$, confirming a higher degree of disorder in MoS$_2$-D$_H$[45]. In addition, both Mo$^{4+}$ 3d$_{5/2}$ and Mo$^{4+}$ 3d$_{3/2}$ doublet peaks in three samples shifted to lower binding energies with increasing sulfur amount, reflecting the reduction of MoS$_2$. This behavior was consistent with decreasing intensity of Mo$^{6+}$ peaks at ~236.2 eV, suggesting the presence of Mo–O bonds in MoS$_2$-D$_H$ decreased when sulfur amount approach to stoichiometric MoS$_2$. These results indicate

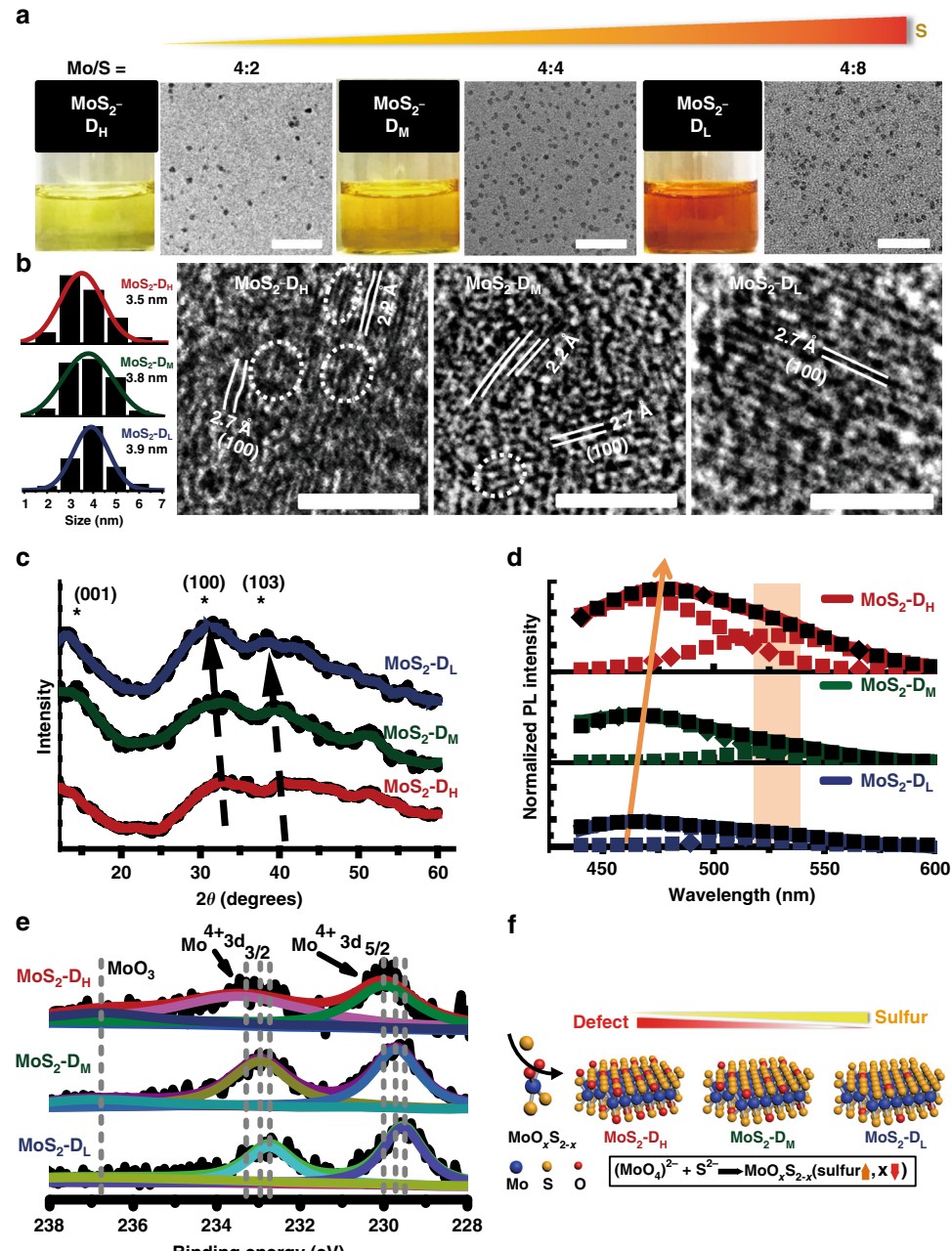

**Fig. 5** Engineering sulfur defects into $MoS_2$ QDs through stoichiometric reaction control. **a** Adjusting precursor ratios produces three kinds of $MoS_2$ suspension in pure water. TEM images show narrow size distribution. Scale bar: 50 nm. **b** Different stoichiometry does not affect overall size of QD $MoS_2$. (Distributions derived from at least $n = 200$). HRTEM images show dislocations and distortions of lattice planes in $MoS_2$ QDs due to intrinsic defects. Scale bar: 2 nm. **c** XRD show gradually enlarged lattice constants with decreasing stoichiometry in three $MoS_2$ QDs. **d** PL spectrum show different emission intensities of the three $MoS_2$ QDs at the same optical absorption. **e** XPS reveal the defect engineering was in the form of $MoO_xS_{2-x}$ by the substitution of oxygen for sulfur. **f** Structural model of the reaction pathway of $MoS_2$ QDs and defect engineering in $MoS_2$ QDs as a function of precursor stoichiometry

that the defect engineering was in the form of $MoO_xS_{2-x}$, by the substitution of oxygen for sulfur within the $MoS_2$ lattices. This is also consistent with the XRD patterns that the substitution of the smaller O atom by the significantly larger S atom in the lattices induce the enlarged lattice constants (Fig. 5c). XPS quantification of the chemical compositions further revealed the atomic ratio of Mo to S varied from 1:1.8 to 1: 2.34 (Supplementary Figure 13). So as sulfur amount increases ($MoS_2$-$D_H$ group to $MoS_2$-$D_L$ group) resulted in the decreasing amount of oxygen atom in $MoO_xS_{2-x}$, giving the direct evidence of sulfur amount-dependent degree of defects in $MoS_2$ QDs.

Collectively the combined studies above, the reaction pathway of $MoS_2$ QDs can be revealed and the defective crystal structures have been proved to obtain via a $MoO_4^{2-} + S^{2-} \rightarrow MoO_xS_{2-x}$ pathway (Fig. 5f). The relatively lower sulfur amount was critically responsible for engineering the $MoS_2$ QDs with S defects. With lower sulfur amount, the reaction became more insufficient, residual Mo−O bonds (originally from $MoO_4^{2-}$) that remained embedded within the otherwise crystalline $MoS_2$ structure; thus creating distortion defects in the observed structure. This reaction behavior is similar to previous observation of controlling synthesis temperature in preparing defective

2D MoS$_2$ nanosheets[39]. Here, 0D MoS$_2$ QDs with controllable defect engineering were realized by simply tuning the synthesis reagent stoichiometry at room temperature.

**Surface vacancy associated singlet oxygen generation.** While the influence of defects on 2D TMDs have been largely studied for tuning optical and physical properties, which widely lead to excellent electronic applications, until now the defect effect on TMDs QDs are suggestive and speculative based on extrapolation from 2D TMDs observations and not applied directly in engineerable defects driven bioapplications. In addition, compared to 2D materials, 0D nanomaterials were supposed to exhibit better biological behavior than their 2D counterparts[46,47]. Thus photochemical related $^1O_2$ generation capacity towards cancer therapy was investigated using our defect tunable MoS$_2$ QDs. The singlet oxygen generation of the MoS$_2$ was examined using 9, 10-anthracenediyl-bis(methylene) dimalonic acid (ABDA) as probe. It was found that without light irradiation, the absorbance of ABDA in three MoS$_2$ QDs solution showed negligible decrease (Supplementary Figure 14). Under the white light irradiation, the absorption intensities within the groups decreased gradually with increased irradiation time (Fig. 6a). In addition, under the same irradiation condition without the presence of MoS$_2$ QDs, both $^1H$-NMR and HPLC results did not show any appreciable change in the spectra of ABDA (Supplementary Figure 15), indicating the degradation of ABDA was indeed induced by the photosensitization from MoS$_2$ QDs. The product of the ABDA trap after MoS$_2$ QDs irradiation was checked by $^1H$- NMR and compared with the corresponding product of Rose Bengal (RB), a positive control known to generate $^1O_2$ under irradiation. Similar chemical shifts were also observed in the corresponding H peaks of the products after irradiation (Supplementary Figure 16), suggesting that the species generated from MoS$_2$ QDs and RB irradiation reacted with ABDA and produced similar ABDA products. We checked again with repeating the irradiation of MoS$_2$-ABDA reaction in D$_2$O or H$_2$O conditions. It was found that there was higher depletion of the ABDA substrate when using D$_2$O while irradiating MoS$_2$ QDs (Supplementary Figure 17). This showed that $^1O_2$ was generated from irradiation of MoS$_2$ QDs. We further checked with a $^1O_2$ 2, 2, 6, 6-tetramethylpiperidine (TEMP) sensor-electron spin resonance spectrum (ESR) assay. We found more product after 5 min of MoS$_2$ irradiation (Supplementary Figure 18). Collectively, $^1O_2$ was likely generated after MoS$_2$ irradiation.

To compare the $^1O_2$ quantum yield of three MoS2 QDs upon light irradiation, calculation was performed based on the photochemical methods reported before[48–50]. The decreased OD at 400 nm (OD$_{400}$) was performed to determine the decomposition rate constants of the photosensitizing process of three MoS$_2$ QDs, which was obtained by fitting the various OD$_{400}$ curves (Fig. 6b). By further integrating the optical absorption of MoS$_2$ QDs in the range 400–800 nm, the $^1O_2$ quantum yield of two kinds of MoS$_2$ QDs relative to MoS$_2$-D$_L$ were determined. The MoS$_2$-D$_H$ and MoS$_2$-D$_M$ groups respectively exhibited $^1O_2$ quantum yield of approximately 2.3 times and 1.7 times of MoS$_2$-D$_L$ QDs' quantum yield (Fig. 6c, Supplementary Figure 19). Physical quenching between $^1O_2$ and N's lone pair electrons of amines may exist; especially those aromatic amines of the BSA-surfactant[51]. It is therefore important to check that the differences in $^1O_2$ quantum yield is not due to different amounts of BSA that are on the surface of the three kinds of MoS$_2$ QDs. We quantified the amount of BSA on the surface of the three kinds of MoS$_2$ using the micro-BCA protein assay. The BCA protein based assay showed no significantly different amounts of BSA on the same amount of the three MoS$_2$ QDs defect types (Supplementary Figure 20). This indicated that the physical quenching due to proteins on the surface of the three MoS$_2$ QDs groups are similar.

Thus, confirming that the significant increase in the $^1O_2$ quantum yields of MoS$_2$-D$_M$ and MoS$_2$-D$_H$ over MoS$_2$-D$_L$ is due to increasing defects.

To further prove the correlations between the degree of defects and photodynamic efficiency, the ROS generation capacity was further detected inside the cancer cells by using dye dichlorohydrofluorescein diacetate (DCFH-DA) as an intracellular ROS indicator. DCFH-DA is non-fluorescent but can be oxidized by ROS to yield highly fluorescent-dichlorofluorescein (DCF). As shown in Fig. 6d, after irradiation, strongest green fluorescence was observed in MoS$_2$-D$_H$ incubated groups, with relatively lower green fluorescence in MoS$_2$-D$_M$ groups and the lowest green fluorescence in MoS$_2$-D$_L$, which clearly confirmed the positive correlation between the degree of defects of MoS$_2$ QDs and ROS generation capacity.

QDs acting as photosensitizers used in photodynamic therapy have attracted great attention in the area of nanomedicine[52–57]. In a typical photosensitizing process, the photosensitizer absorbs a quantum of light from ground state (S$_0$) to the excited singlet state (S$_1$), which is further converted to the excited triplet state (T$_1$) via intersystem crossing (ISC). The subsequent energy transferring from T$_1$ to triplet oxygen ($^3O_2$) result in the formation of singlet oxygen ($^1O_2$)[58]. The quantum yield of $^1O_2$ generated by photosensitizers can be expressed as equation (1)[59]

$$\phi_\Delta = \phi_T \phi_{en}, \tag{1}$$

where $\Phi_T$ is the quantum yield of T$_1$ formation and $\phi_{en}$ is the efficiency of energy transfer from T$_1$ to $^3O_2$. To understand the enhanced ROS generation mechanism in cells, we analyzed the relationship between the defects and corresponding ROS generation behavior in MoS$_2$ QDs. Previous records have confirmed that $\Phi_T$ is determinate by the ISC rate constants $k_{ISC}$, which could be estimated from Eq. (2)[60]

$$k_{ISC} \propto \frac{(T_1|H_{SO}|S_1)^2}{\Delta E_{S1-T1}} \tag{2}$$

Here $H_{SO}$ represents the Hamiltonian for the spin-orbit perturbations and $\Delta E_{S1-T1}(\Delta E_{ST})$ is the energy gap between S$_1$ and T$_1$. Therefore by engineering the HOMO-LUMO energy level in molecules to reduce the energy gap $\Delta E_{ST}$, high photosensitizing efficiency could be achieved[61,62]. In our MoS$_2$ QDs experiments, we speculated that the sulfur vacancy derived defects engineered the bandgap of MoS$_2$ QDs, thus modulating the $\Delta E_{ST}$ to affect the photosensitizing capacity. To clarify this hypothesis, theoretical simulation using DFT was performed to check the density of states (DOS) of MoS$_2$ QDs. After constructing a $3 \times 3 \times 1$ supercell structure, zero, one and two sulfur atoms in the supercell were removed and substituted by oxygen atoms (Fig. 6e). This leads to different degrees of sulfur defects in MoS$_2$, which denoted as pristine MoS$_2$, MoS$_2$-1$D_0$ and MoS$_2$-2$D_0$. The calculated DOS in Fig. 6f demonstrated that the vacancies of sulfur atoms and the substitution of oxygen atoms act as n-type dopants. The more substitution of more electronegative oxygen atoms led to more charge carriers, with narrower bandgap from pristine MoS$_2$ of 1.68 eV to 1.42 eV (MoS$_2$-1$D_0$) and 1.30 eV (MoS$_2$-2$D_0$). This decreasing bandgap of the simulated defective mode in MoS$_2$ QDs qualitatively agrees with experimental observations from optical band gap (Supplementary Figure 21). Our calculation indicate that the sulfur defects in MoS$_2$ QDs tend to reduce the bandgap with lowering $\Delta E_{ST}$, which improved ISC efficiency and account for the observed enhanced $^1O_2$ generation in the photosensitizing process.

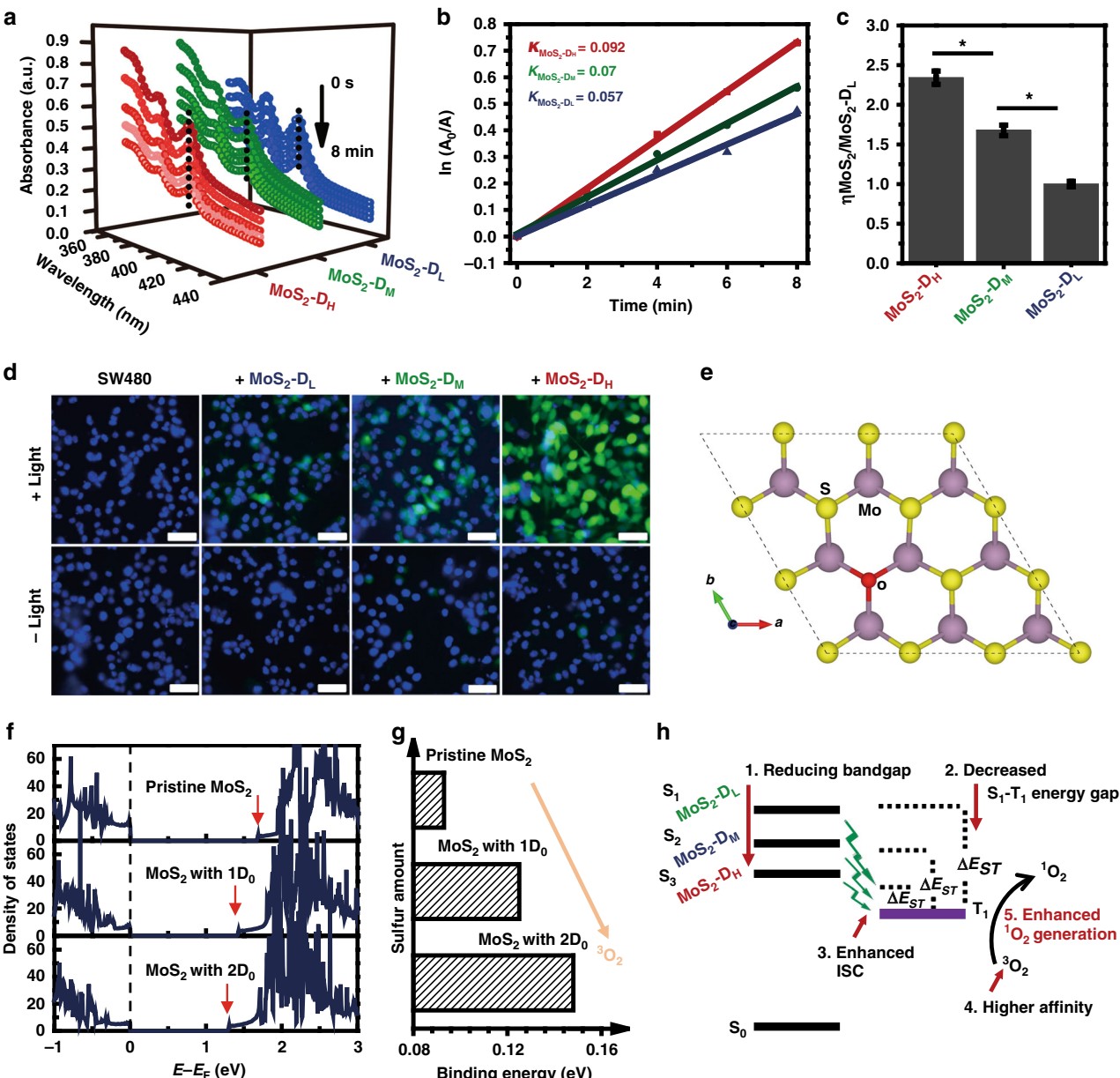

**Fig. 6** Positive correlation between sulfur defects and photodynamic efficiency in QDs. **a** Absorption spectra of three kinds of MoS$_2$ QDs in the presence of ABDA under light irradiation (0.1 W cm-2, 8 min). **b** Typical decomposition rate of the photosensitizing process, where $A_0$ is the absorbance of initial absorbance of ABDA and $A$ is the absorbance of ABDA under light irradiation at different time points (Full data plots can be found as Supplementary Figure 19). **c** Relative $^1O_2$ quantum yield of two MoS$_2$ QDs groups relative to MoS$_2$-D$_L$ QDs. Mean ± SD, $n = 4$, Student's $t$-test, $p^* < 0.05$. **d** Detection of ROS generation at cellular level in colon cancer cell line, SW480 by intracellular ROS indicator H$_2$DCFH-DA. Scale bar: 50 μm. **e** Structure illustration for the substitution of oxygen for sulfur within MoS$_2$ lattices. **f** Calculated density of states of MoS$_2$ QDs show the defects reduce the bandgap. **g** More sulfur defects induce stronger binding affinity of $^3O_2$ to MoS$_2$. **h** Proposed defect related $^1O_2$ generation mechanism of MoS$_2$ QDs

On the other hand, the substitution of sulfur atom with the more electronegative oxygen atom modulated the charge density distributions in MoS$_2$ crystals, affecting the Gibbs free energy for $^3O_2$ adsorption. To understand the binding affinity after different degrees of defect, the binding energies of $^3O_2$ on three MoS$_2$ QDs were calculated by DFT. As shown in Fig. 6g, the binding energies for oxygen adsorption on MoS$_2$ QDs decrease with less sulfur defect. The binding energies of $^3O_2$ on MoS$_2$-2$D_0$ are 0.15 eV, which are much larger than MoS$_2$-1$D_0$ (0.13 eV) and pristine MoS$_2$ (0.09 eV). Moreover, with three sulfur atoms substituted by oxygen atoms, binding energy further increased to 0.17 eV. This result suggested that a higher defect MoS$_2$ QD possessed a

stronger binding affinity toward $^3O_2$ adsorption, which may allow higher oxygen coverage on the surface of MoS$_2$ QD in the energy transfer process, thus paving the way for higher $\phi_{en}$ in $^1O_2$ generation.

Besides the reduced bandgap and the strengthened binding affinity between MoS$_2$ and $^3O_2$ (Fig. 6h), the spin-orbit perturbations ($H_{SO}$) in defective MoS$_2$ QDs could also affect the intersystem crossing in the photosensitizing process. The vibronic coupling involved in Mo–S and Mo–O bonds would be significantly increased due to the increasing degree of defects. The decrease in size increases edge sites and dangling bonds in MoS$_2$ QDs. This further enhances the likelihood of vibrational modes

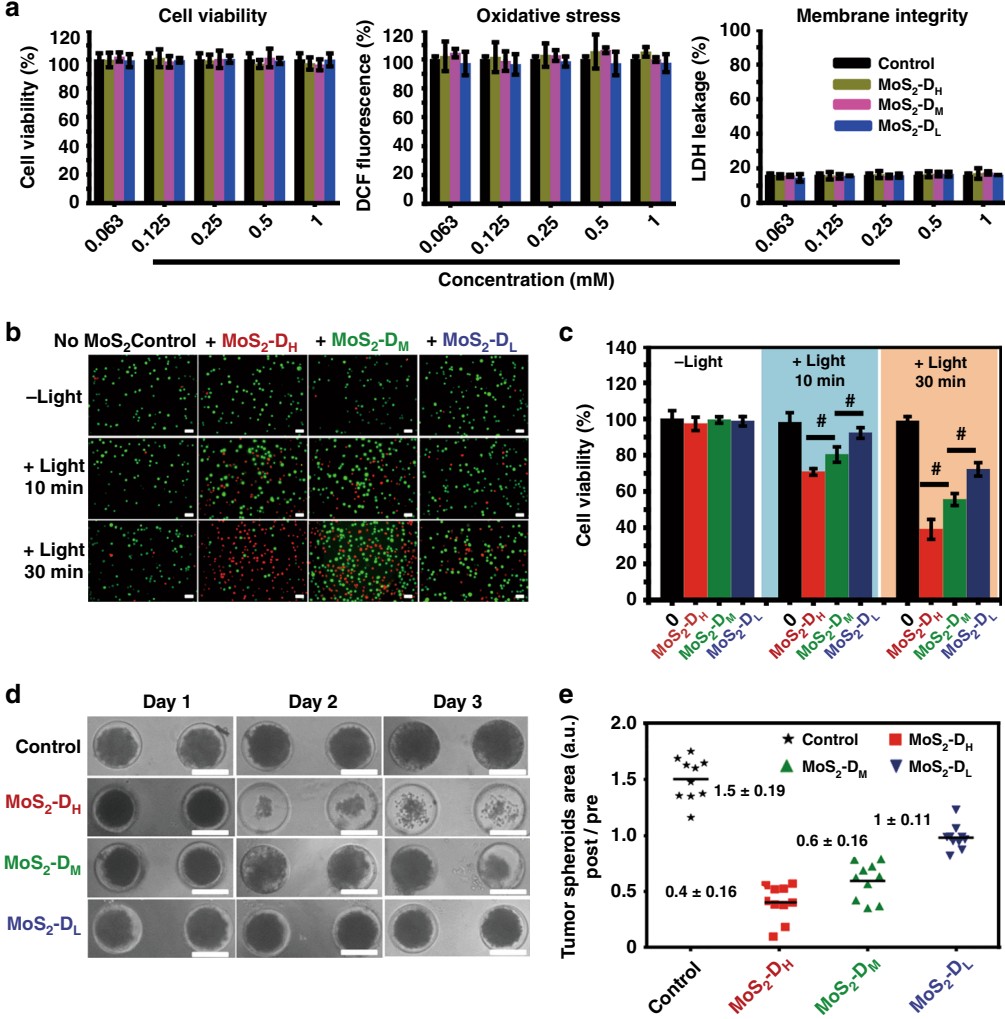

**Fig. 7** MoS$_2$ QDs with more S defects produce more oxidative stress in cancer cells. **a** MoS$_2$ QDs treatment (without light) on human endothelial cells (HMVEC) showed negligible cytotoxicity, negligible excessive ROS generation and low apoptosis induction. The measurements were performed in triplicate. **b** Calcein AM and PI co-staining and **c** cell viabilities of SW480 cells incubated with different defect laden MoS$_2$ QDs under white light exposure. The measurements were performed in triplicate. **d** MoS$_2$-D$_H$ showed the highest size reduction of 3D tumor spheroids assay. This implied high penetration of QDs into the interior of the 3D tumor spheroids mass with the highest defect group producing the highest oxidative stress that killed the cells. **e** Quantification of 3D tumor spheroids area post treatment and laser excitation with MoS$_2$-D$_H$ showing the greatest cell death amongst the three MoS$_2$ groups. Mean ± SD, $n = 10$, Student's $t$ test, $p^{\#} < 0.05$. Scale bar: 100 μm

hence promoting the intersystem crossing. The real exciton splitting situation due to the presence of defects and quantum confinement effect of typical semiconductor nanocrystals behavior in MoS$_2$ QDs' case complicated their energy state. Here the proposed singlet and triplet states of MoS$_2$ QD structure models was under the consideration of surface molecular states of MoS$_2$ QDs. Further investigations to understand the electronic structure and exciton of MoS$_2$ QDs are warranted.

**Defect-dependent photodynamic therapy to kill cancer cells.** Inspired by defect-dependent ROS generation capacity, the photodynamic efficiency of the three MoS$_2$ QDs for killing cancer cells were further conducted. The material biocompatibility was first investigated. Cell viability, LDH and ROS level were examined to systematically evaluate the cytotoxicity effect (biocompatibility) of three MoS$_2$ QDs using endothelial cells as a model for a non-cancer cell type which would be in contact with introduced MoS$_2$ QDs bioapplications. Fig. 7a suggest that no obvious toxicity of all the three MoS$_2$ QDs on human

microvascular endothelial cell (HMVEC) cells at MoS$_2$ QDs concentration as high as 1 mM for 24 h, suggesting good biocompatibility of all three MoS$_2$ QDs types. There was no additional upregulation of ROS as the MoS$_2$ QDs was not irradiated with light. When increasing the dose of MoS$_2$ QDs, there was no increasing LDH leakage due to loss of membrane integrity as a result of induced apoptosis. We next used MoS$_2$ QDs as photosensitizers in photodynamic cancer cell therapy in an in vitro model. Using SW480 cells as model cancer cells, Calcein AM (green fluorescence) and propidium iodide (PI, red fluorescence) staining were performed to visualize live/dead cells in the cancer cell therapy process. Control cell group with light treatments only or MoS$_2$ QDs incubation only (without light treatment) maintained high cell viability (Fig. 7b). As expected, cells incubated with higher defects MoS$_2$-D$_H$ QDs showed considerably lower cell viability under light irradiation compared to MoS$_2$-D$_M$ and MoS$_2$-D$_L$ treatment groups (Fig. 7c). After being irradiated for a short 10 min, only 72% of cells incubated with MoS$_2$-D$_H$ were viable, which further decreased to 39.7% when the irradiation

time prolonged to 30 min. At the same treatment concentrations, cell viability of MoS$_2$-D$_M$ and MoS$_2$-D$_L$ groups decreased from 81% to 55.2 and 92% to 72% respectively, as the irradiation time was prolonged from 10 min to 30 min.

These results demonstrated that MoS$_2$ QDs with higher defects are more effective photodynamic agents to kill cancer cells over those with lower defects at the same concentration.

To further test the photodynamic efficiency in vitro on cancer cells, 3D SW480 colorectal tumor spheroids were used as a more realistic cancer model to simulate the therapeutic responses of three MoS$_2$ QDs groups[63,64]. To examine the photodynamic effect, 3D tumor spheroids were incubated with different MoS$_2$ QDs groups for 6 h to allow for increased penetration and then irradiated with white light for 30 min. After three days, as expected, the size of tumor spheroids incubated with MoS$_2$ QDs with a single irradiation show higher degree of growth retardation compared to control group without MoS$_2$ treatment (Fig. 7d, e, Supplementary Figure 22). More importantly, the MoS$_2$-D$_H$ treated groups show the greatest tumor spheroids size reduction, as compared with that of the MoS$_2$-D$_M$ and MoS$_2$-D$_L$ groups at equivalent MoS$_2$ concentrations. These results indicated that the degree of defects was related to the photodynamic efficiency of MoS$_2$ QDs, showing the possible defect engineering in QDs to enhance their therapy effect while reducing their actual dosage used in clinic, thereby preventing the potential chronic side effects and complications.

## Discussion

In summary, we have demonstrated the biomineralization assisted bottom-up strategy for synthesizing a wide range of QDs from the TMD family with using mild conditions. The strategy employed straightforward chemical reactions between sodium chalcogenides and transition metal chlorides or transition metal oxides with a high yield reaction within a matter of tens of seconds. Disulfide bonds were shown to be the functional group for size quality control, and further verified by DFT simulations. The major benefit of the synthetic route is the ability of controllable defects engineering on TMD QDs via tuning precursor stoichiometry. Our studies revealed that the reaction pathway of MoS$_2$ QDs and the defective crystal structures might be $(MoO_4)^{2-} + S^{2-} \rightarrow MoO_xS_{2-x}$. This synthetic strategy simplifies the synthetic process with mild conditions, enriching the TMD QDs library for exploring their physical and chemical properties and related applications. Our bottom-up strategy for synthesizing TMDs QDs provided an ideal platform for investigating the correlation between the surface defects of semiconductor nanomaterials and the corresponding capacity of anti-cancer oxidative stress generation. Strongly positive correlation between degrees of sulfur defects and photodynamic efficiencies could be observed in the prepared MoS$_2$ QDs. The density of states and molecular dynamics calculations suggest that the sulfur defect in MoS$_2$ QDs reduced the bandgap and strengthened the binding affinity between MoS$_2$ QDs and $^3O_2$. This may have contributed to the intersystem crossing and energy transfer separately in the photosensitizing process; highlighting the significant potential for defect engineering as an intrinsic alteration tool used in conjunction with existing TMDs bionanotechnologies[65–67] without adding yet another material.

## Methods

**Materials**. All chemicals and reagents were used as received without any further purification. MoCl$_5$ (95%), MoO$_3$ (95%) and Bovine Serum Album (BSA) were purchased from Sigma-Aldrich Co. (USA). Deionized water was used throughout the synthesis.

**Sample characterization**. The morphologies and size of MoS$_2$ QDs samples were characterized by TEM (FE-TEM; JEOL JEM-2100F, Japan). Nanoparticle sizes were determined by measuring no <200 randomly selected nanoparticles from TEM micrographs with ImageJ (http: //rsbweb.nib.gov/ij/). The powder XRD measurements were performed using a Bruker D8 advanced diffractometer with a Cu Ka irradiation in the 2θ range of 20$^0$-60$^0$. The elemental composition and binding energy of the sample were characterized by X-ray photoelectron spectroscopy (XPS; AXIS HIS, Kratos Analytical). The absorbance spectrum scanning and fluorescence intensities were conducted using microplate reader (BioTEK H4FM, USA). Fluorescence imaging were taken with inverted fluorescence microscope (Leica DMI6000, Germany) and the phase contrast images were captured by inverted microscope (Olympus-CX41, Japan).

**Synthesis of MoS$_2$ quantum dots**. In a typical procedure, Mo-precursor solution was firstly prepared by dissolving MoCl$_5$ or MoO$_3$ into dH$_2$O by adjusting the pH value to be 11. The resultant solution was transparent and colorless after sonication. Then 1 mL of Mo precursor solution mixed into 39 mL of BSA solution (1 mg/mL) followed by adding 0.2 mL 0.5 M Na$_2$S under vigorous stirring at room temperature. The pH value of the mixed solution was subsequently adjusted to 6~7 by adding 1 M HCl. After neutralizing the pH, a clear yellow solution of MoS$_2$ quantum dots was produced quickly.

**Synthesis of other TMD quantum dots with BSA**. The protocol of synthesizing other TMD quantum dots was similar to MoS$_2$ quantum dots. The only difference was the preparation of Se- and Te- precursors. In a typical preparation of Se precursor, 79.9 mg of Se powder was added into 1 mL of NaBH$_4$ solution (80 mg) to reduce it at ambient condition. 30 min later, black Se powder disappeared completely and the clear solution was Se precursor. For the preparation of Te precursor, 31.9 mg of Te powder was added into 200 µL NaBH$_4$ solution (28.4 mg). After 30 min, black Te powder fully disappeared and the clear solution was Te precursor. For the synthesis of WS$_2$, RuS$_2$, MoTe$_2$, WSe$_2$ and RuSe$_2$, similarly mild conditions were carried as MoS$_2$ above.

**Photodynamic evaluation of three MoS$_2$ QDs with ABDA**. In the experiments, ABDA as the probe of $^1O_2$ was added to different MoS$_2$ QDs dispersion solution with final concentration of 10 µM. Then the mixture solution was transferred to a cuvette and exposed to white light (400–800 nm, 100 mW) perpendicularly for 8 min. The light source irradiated a region of 1 cm$^2$. For the convenience of observation, the OD of three MoS$_2$ QDs at 400 nm was adjusted to the same. The absorbance change at 400 nm was recorded at various time points to obtain the degradation rate of ABDA. For the comparison of $^1O_2$ quantum yield, all the $^1O_2$ quantum yield of MoS$_2$ was normalized to that of MoS$_2$-D$_L$, relative $^1O_2$ η to MoS$_2$-D$_L$ using the following equation:

$$\frac{MoS_2}{MoS_2 - D_L} = \frac{K_{MoS_2} A_{MoS_2-D_L}}{K_{MoS_2-D_L} A_{MoS_2}},$$

where $K_{MoS_2}$ and $K_{MoS_2-D_L}$ represent the decomposition rate constants of the photodegradation of ABDA with different MoS$_2$, which were determined by plotting Log$_e$(Abs$_0$/Abs) versus irradiation, where Abs$_0$ is the initial absorbance of ABDA, Abs is the ABDA absorbance at different irradiation time. $A_{MoS_2}$ and $A_{MoS_2-D_L}$ refer to the light absorbed by different MoS$_2$ and MoS$_2$-D$_L$ respectively, which are determined by integration of the optical absorption bands in the wavelength range from 400 to 800 nm.

**Intracellular ROS assay**. DCF-DA probed the ROS generation inside cells. SW480 cells were cultured on 8-well cell chamber slides overnight. Overnight medium was replaced with fresh medium containing different MoS$_2$ QDs (at 0.5 mM). After incubation for 6 h, DCF-DA (25 µM) was added for 15 min, and then either left in the dark or irradiation with white light (100 mW cm$^{-2}$) for 5 min. After washing and staining the cells with Hoechst 33342 (5 µg/mL), fluorescence images of the cells were captured using Nikon A1 confocal microscope (Nikon, Japan).

**Photodynamic killing of SW480 cancer cells**. To assess the photodynamic effect of MoS$_2$ QDs on SW480 cells, SW480 cells was seeded and cultured in 96-well plates overnight. Then different groups of MoS$_2$ QDs were used to treat SW480 cells for 6 h. Subsequently, cells were irradiated with white light (100 mW cm$^{-2}$) for 10 and 30 min. After treatments, the cells were further cultured for 12 h. Cell viability after different treatments was evaluated by WST assays and Tali Image based Cytometer (Life Technologies, USA).

**Formation and photodynamic efficiency in SW480 colorectal tumor spheroids**. To examine the photodynamic effect on 3D SW480 colorectal tumor spheroids, 3D cell spheroids were first prepared. The solidified agrose micro-molds were sterilized by UV irradiation for 30 min and then equilibrated with DMEM medium for 12 h. After that 50 µL of SW480 cell suspension were seeded into each agarose micro-mold, 5 min after the cells settle down into the micro-mold, 500 µL of DMEM medium was added to the well. After 24 h, 3D SW480 cell spheroids were

formed due to gravity and the aggregation of cells. For photodynamic therapy treatment, the different $MoS_2$ QDs were added into 3D SW480 cell spheroids. After 6 h, 3D SW480 cell spheroids were either exposed to dark or white light (100 mW cm$^{-2}$) for 30 min. The images of 3D SW480 cell spheroids were captured by a light microscope (Olympus-CX41, Japan) in the subsequent days and the size of the 3D SW480 cell spheroids were measured with ImageJ software.

**Density functional theory (DFT) binding energy siumlation**. Adsorptions of various chemical groups on $MoS_2$ were investigated by performing simulations at the density functional theory (DFT) level as implemented in the Vienna ab initio simulation package (VASP)[68], with the exchange-correlation functional of Perdew-Burke-Ernzerhof (PBE)[69]. The long-range van der Waals interactions were calculated within the Tkatchenko and Scheffler scheme to avoid the empirical parameters[70], while the self-consistent screening and polarizability contraction effects were also taken into account, in view of their important roles in determining the weak inter-molecular interactions[71]. The $MoS_2$ layer was modeled with a vacuum region of 20 Å. The first Brillouin zone was sampled with a $k$-point mesh with a plane-wave cutoff of 450 eV. The various molecules including $CH_3$–X (X = –OH, –SH, –$NH_2$, –COOH, –$SSCH_3$ and –$C_6H_5$) and $CH_3$–CONH–$CH_3$ were simulated following the equation $E_{ad} = E_{MoS_2+group} - E_{group} - E_{MoS_2}$ with the results summarized in Supplementary Table 1.

**Methodology for density of states (DOS) of models $MoS_2$.**

- DFT based VASP.
- PAW-PBE potential electron-ion interaction and exchange-correlation.
- $3 \times 3 \times 1$ supercell structure is applied
- 500 eV for plane-wave expansion cutoff.
- $7 \times 7 \times 1$ Gama-centred k-point mesh.
- All structures were relaxed until the force is smaller than 0.001 eV/Angstrom with a total energy convergence criterion of $1 \times 10^{-6}$ eV.
- Vacuum is 15 Å.
- The relaxed lattice parameter for $MoS_2$ monolayer is 3.190 Å

## Data availability
All the relevant data are available from the corresponding authors upon reasonable request.

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

## Acknowledgements

We acknowledge the funding provided by the National Research Foundation, Prime Minister's Office, Singapore, under Competitive Research Program (Award No. NRF-CRP13-2014-03).

## Author contributions

X.D. and D.T.L. conceived the project, the hypotheses and the experiments. X.D., F.P., W.B.G., Z.J. performed the experiments. X.D., C.T.L. and D.T.L. analyzed the results. X.D., F.P., G.S., K.P.L., C.T.L., and D.T.L. discussed the results. X.D., C.T. L. and D.T.L. wrote the manuscript.

## Additional information

**Competing interests:** The authors declare no competing interests.

**Journal Peer Review Information:** *Nature Communications* thanks the anonymous reviewers for their contributions to the peer review of this work. [Peer reviewer reports are available].

