## [Peer Review File · Nature Communications]

Reviewers' comments:

Reviewer #1 (Remarks to the Author):

This manuscript consists of two parts, namely an excellent new synthetic method for MoS₂ Quantum Dots, and an investigation into the QD surface and surface defects, and the inlet oxygen sensitizing ability of these QDs as a function of surface defects.

The first part (QD synthesis) looks fine, although by itself the impact would probably not be broad enough to justify publication in Nature Communications.

The second part of this work is really interesting, and might justify publication in this journal, although major revision is needed. A number of control experiments and some text revisions need to be undertaken before publication can be considered:

The authors used white light (400 to 800 nm) to irradiate mixtures of the MoS₂ QDs and a singlet oxygen trap, namely 9,10-anthracenediyl-bis(methylene) dimalonate (ABDA). However, this compound absorbs strongly near 400 nm as well. Hence when the authors irradiate the mixture, they will excite BOTH the QDs and the anthracene derivative! Itself! The disappearance of the ABDA could indeed be due to [4+2] cycloaddition of singlet O₂ at the central ring, but it could also be due to other photochemical reactions of the excited ABDA. Furthermore, since the process was only followed by UV/vis, all we know is that the ABDA UV/vis peaks disappear - we do not know if the 9,10-endoperoxide or some other product is formed. The authors should check endoperoxide product formation by ¹H NMR at least. There could also be Type I (radical) type photooxidation processes which do not involve singlet O₂. It may well be that all of the disappearance of the ABDA is due to ¹O₂, but the following control experiments are needed to establish this: What happens if ABDA is excited in the presence of O₂ (same light source) but no Mo QDs? To differentiate between Type I and Type II processes, the authors could either look at solvent effects (¹O₂ has a much longer lifetime in deuterated solvents, and hence ABDA disappearance would be faster in such solvents) or the effect of physical singlet oxygen quenchers. Finally, the method employed for the relative singlet oxygen quantum yields has one more disadvantage, namely an a priori assumption that the different Mo QDs do not physically quench ¹O₂, or, if they do, that they do so at the same rate. (Physical quenching would decrease the rate of ABDA disappearance). The authors should at least discuss this.

Other comments:

"The quantum yield of 3O_2 generated by photosensitizes..." I would guess that the authors mean "The quantum yield of 1O_2 generated by photosensitizes..."

The authors seem to talk about ROS and 1O_2 as if these were the same thing. ROS include 1O_2 , but also includes radicals generated by Type I processes. The authors only talk about 1O_2 when they mean singlet oxygen and use ROS only when they mean singlet oxygen and free radical type processes.

Page 5, line 106:

"Subsequent adjusting of pH by adding HCl activates the sulfur precursors..."

What was the pH adjusted to? Please be specific and give a value.

Page 5 lines 110-111:

"This ease paves the way for higher scalability..." Do the authors mean "This easily paves the way for higher scalability..."?

Overall, this is a very nice paper that goes far beyond the typical phenomenological descriptions found in most papers on QD synthesis and properties. If the control experiments outlined above confirm the author's hypothesis that surface defects increase the 1O_2 quantum yield of their QDs, this paper could be publishable in Nature Communications.

Reviewer #2 (Remarks to the Author):

The manuscript "Library Construction of Defect Variable Bioactive Transition Metals Dichalcogenides Quantum Dots" describes a biomineralization assisted bottom-up strategy for the synthesis of a wide library of transition metal dichalcogenides (TMD) QDs. The reactions they used are very fast (10-20 s) with very soft conditions of room temperature, aqueous and atmosphere. Further study on MoS₂ QDs synthesized by the method for biomedical applications demonstrated that increased sulfur defects correlated well with increased oxidative stress generation from photodynamic effect in cancer cells. One of the most exciting advantages of the proposed method is the preparation of atomic surface defects of QDs, which allows researchers to investigate the nanoparticles defect effects on their functions when using in different field, because the atom defect of nanosurface is an

open question in many fields. So, I recommend to publish this work at Nature Communication after minor revision.

The minor questions:

1. Could the absorbance band of MoS₂ be tuned to the optimized NIR light-transparent window of biomedical applications?
2. Figure 6a, how long were HMVEC cells incubated with MoS₂ to evaluate the viability? What about the cell viability of SW480 after treatment of MoS₂?
3. What about the cellular uptake and subcellular localization of MoS₂ in HMVEC and SW480?
4. Please comment of the details of the laser such as beam diameter, energy distribution in the beam, total power, etc. and how the irradiation experiments were performed (for instance, a region of xx cm² was irradiated). It is important for reproducibility.
5. What's the average diameter of the 3D spheroids? Could the laser spot cover the spheroid?
6. PDT is mainly applied to the superficial treatments, such as breast cancer and skin diseases. Please describe the experiment of MoS₂-based PDT in the in vivo model of SW480 colon cancer?

Reviewer #3 (Remarks to the Author):

This manuscript describe the preparation of TMD QDs at room temperature in 10-20 s by the reaction of sodium chalcogenides with metal chlorides or oxides in BSA as surfactant. The authors also demonstrated the control over defects by using different ratios of chalcogen and metal. The defects are created by the inclusion of oxygen which replaces sulphur in the crystal lattice. Pristine and the defect samples were then investigated for anticancer oxidative stress generation. A strong correlation was shown between the degree of sulphur defects and photodynamic efficiency.

This work is highly significant as it provides a simple and quick method in aqueous solution for the preparation of TMD QDs which can be applied for the whole TMD family. The creation and the control over the defects and the relationship of the defects to their photodynamic efficiency is an important observation.

I am satisfied that all the results presented in this work are verified by the experimental evidence. The work is presented clearly with sufficient experimental detail and good discussion. I recommend its publication without any change.

Reviewer #4 (Remarks to the Author):

This work reports on the bottom-up synthesis of transition metal dichalcogenide (TMDC) quantum dots (QDs), the role of defects in QDs on the production of reactive oxygen species, and the efficacy of these QDs to treat cancer cells. While the authors performed a large number of experiments in this work, the science would be better communicated if only the most important results were presented with greater detail in their analysis. In general, there are two narratives to this study: (i) the facile synthesis of TMDC QDs and (ii) the use of MoS₂ QDs for cancer treatment. While the facile synthesis of MoS₂ QDs (that are more monodisperse than those obtained through top-down methods) is an important development, questions (as outlined below) remain unanswered about the second part of this study. These questions lessen the potential impact of the manuscript.

Some specific suggestions/questions:

1. A more expansive literature review would be helpful to understand the current status of bottom-up MoS₂ QD synthesis.
2. Which phase are the MoS₂ QDs?
3. Regarding XPS quantification of chemical compositions, if the sulfurs on cysteine ligands are obscuring the Mo:S ratio, does the ratio change with the other ligands used in the study?
4. How would surface states contribute to the calculated band gap of 3.45 eV (as the authors mentioned around lines 144-146)?
5. The authors mention that they believe that one MoS₂ QD is interacting with one BSA molecule from DLS measurements. Have the authors tried to use NMR to better quantify the degree of ligand-binding to the surface of the QD?
6. Does the number of ligands present on the surface of the QDs change with defect density (or amount of sulfur)?

7. Does the photodynamic therapy selectively kill cancer cells? Did it kill the HMVEC cells as well?
8. Why were SW480 cells used as opposed to other cancer cell lines?
9. Have similar studies on cancer cells been done with other MoS₂ structures (nanodisks, nanosheets)?
10. The analysis of photoluminescence measurements (278-300) with regard to defects seems incomplete. The authors state that the trend in photoluminescence quantum yield is likely due to different surface defect states, and that the relatively higher quantum yield of MoS₂-DH QDs suggests they are the most defective. While there are instances where defects may enhance quantum yield, they are not necessarily directly related in this way, and are often times inversely related. The authors acknowledge that two possible emission peaks present in the spectrum may be attributed to “intrinsic state emission (electron-hole recombination) and defect state emission”, but no deconvolution of the peaks is done. By visual inspection, the PL from the intrinsic state emission is also significantly higher in the MoS₂-DH QDs (Figure 4d), the most defective and brightest QDs, and not just the possible defect emission peak.
11. The authors’ text (Line 68) and Scheme 1 claim a universal bottom-up route to synthesis of TMD quantum dots. While seven compounds are prepared, this is not sufficient to claim a universal approach as indicated in the scheme.
12. Of the other TMD nanodots, the authors state “highly homogenous of size distribution with diameters below 10 nm” (Line 223), and histograms in Figure 3 are cut off at 7-10 nm. In the corresponding TEM images, there are particles of greater size than 10 nm based on the 100 nm scale bar. The authors note that some rod-like shape can be found in MoTe₂ samples, but if this is in reference to the larger particles, they are present in images of other TMD species as well. Are these particle aggregates? Additionally, in Figure 1b, it looks like the standard deviation of the QDs is around 25%. Does the Gaussian fit change with smaller bin sizes (0.25 nm as opposed to 1 nm)?
13. In the discussion on the photosensitization by TMD NDs, beginning on Line 374, the authors argue, “sulfur defects in MoS₂ QDs tend to reduce the bandgap with lowering Φ_{EST} ”. A reduced bandgap doesn’t necessarily lead to a smaller energetic splitting between the singlet and triplet states, and the splitting is only one factor determining intersystem crossing rate. Additionally, “singlet” and “triplet” states of semiconductor nanocrystals are not easily differentiable.
14. Line 187: The authors state “FWHM value” when giving the mean particle size.

The reviewer's comments are italicized.

Our responses to the reviewer's comments are in normal black fonts.

Our revisions in the manuscript are in green highlights.

Reviewer #1 (Remarks to the Author):

This manuscript consists of two parts, namely an excellent new synthetic method for MoS₂ Quantum Dots, and an investigation into the QD surface and surface defects, and the inlet oxygen sensitizing ability of these QDs as a function of surface defects.

The first part (QD synthesis) looks fine, although by itself the impact would probably not be broad enough to justify publication in Nature Communications.

Responses: Thank you for your nice comment on our synthesis methodology.

The second part of this work is really interesting, and might justify publication in this journal, although major revision is needed. A number of control experiments and some text revisions need to be undertaken before publication can be considered:

Responses: Thank you for your nice comment on our defect studies and findings. We have undertaken several experiments to address the concerns. Please refer to Figure S15.

Figure S15. $^1\text{H-NMR}$ (a) and HPLC (b) spectrums of ABDA before and after irradiation for 8 mins.

The authors used white light (400 to 800 nm) to irradiate mixtures of the MoS₂ QDs and a singlet oxygen trap, namely 9,10-anthracenediyl-bis(methylene) dimalonic acid (ABDA). However, this compound absorbs strongly near 400 nm as well. Hence when the authors irradiate the mixture, they will excite BOTH the QDs and the anthracene derivative! Itself! The disappearance of the ABDA could indeed be due to [4+2] cycloaddition of singlet O₂ at the central ring, but it could also be due to other photochemical reactions of the excited ABDA. Furthermore, since the process was only followed by UV/vis, all we know is that the ABDA Uv/vis peaks disappear - we do not know if the 9,10-endoperoxide or some other product is formed. The authors should check endoperoxide product formation by ¹H NMR at least. There could also be Type I (radical) type photooxidation processes which do not involve singlet O₂. It may well be that all of the disappearance of the ABDA is due to ¹O₂, but the following control experiments are needed to establish this: What happens if ABDA is excited in the presence of O₂ (same light source) but no Mo QDs?

Responses: Thank you for your nice comment on our defect studies and findings. We have undertaken a series of experiments to address the concerns (Figure S15). To rule out the possibility of the disappearance of the ABDA was due to the photochemical reactions of the excited ABDA that not related to singlet O₂. We follow the suggestion of employing ¹H NMR to check the product of ABDA, which was excited in the presence of O₂ but no Mo QDs with the same irradiation condition (0.1 W/cm², 8 min). Due to the relatively low detection sensitivity of NMR, here the concentration of ABDA was 1000 times of the real usage amount (10 mM). Nonetheless, no appreciable peak shift was detected from ¹H NMR with the product with and without irradiation. We further analyzed the product of the real used concentration of ABDA (10 μM) under irradiation. The HPLC result was consistent with the ¹H NMR result that no obvious change was observed compared to the pristine ABDA in the spectrums. Given the MoS₂ does not affect ABDA in the absence of irradiation (Figure S14), the ¹H NMR combined with HPLC results confirm that the disappearance of the ABDA was indeed due to the ¹O₂ generated by MoS₂ QDs under light irradiation and finally resulting in the decrease of ABDA absorbance.

To differentiate between Type I and Type II processes, the authors could either look at solvent effects (¹O₂ has a much longer lifetime in deuterated solvents, and hence ABDA disappearance would be faster in such solvents) or the effect of physical singlet oxygen quenchers.

Response: While Type II photochemical process involves energy transfer between excited T₁ states of photosensitizers and ³O₂ to generate cytotoxic ¹O₂, which is the characteristic of Type II photodynamic reaction, Type I photochemical process occurs directly between T₁ photosensitizer and other molecules to form radical cation and radical anion. In our study the ABDA molecules were specially chosen for probing ¹O₂. As anthracene derivative, ABDA has been widely used as ¹O₂-specific probe acting as both physical and chemical quencher for ¹O₂, which typically generate in type II photodynamic reaction. (Mooi, Sara M., and Belinda Heyne. "Amplified production of singlet oxygen in aqueous solution using metal enhancement effects." *Photochemistry and photobiology* 90.1 (2014): 85-91. Kuznetsova, N. A., et al. "New reagents for determination of the quantum efficiency of singlet oxygen generation in aqueous media." *Russian Journal of General Chemistry* 71.1 (2001): 36-41. Idris, Niagara Muhammad, et al. "In vivo photodynamic therapy using upconversion nanoparticles as remote-controlled nanotransducers." *Nature medicine* 18.10 (2012): 1580.). Combined the experiment and analysis above we believe the involved photooxidation was mainly due to Type II process.

Finally, the method employed for the relative singlet oxygen quantum yields has one more disadvantage, namely an a priori assumption that the different Mo QDs do not physically quench ¹O₂, or, if they do, that they do so at the same rate. (Physical quenching would decrease the rate of ABDA disappearance). The authors should at least discuss this.

Responses: Thank you for pointing out that. Yes, the method employed for the calculation is

mainly refer to external quantum yield and is relative. It is the combining results of internal quantum efficiency and quenching effect from MoS₂. Several studies have shown that physical quenching exist between ¹O₂ and N-lone electron pair of amines, especially aromatic amine which is the consisting component of amino acid in BSA (Matheson, I. B. C., et al. The quenching of singlet oxygen by amino acids and proteins. *Photochem. Photobiol.* **21**, 165-171 (1975).). Hence, here the surfactant BSA was considered to be the mainly physical quenching effect in MoS₂ QDs. On the other hand, the number of surfactant molecules on the surface of particles was supposed to be mainly affected by particle size, which determine the surface coverage of surfactant molecules on the nanocrystal surface. Since all three MoS₂ QDs have similar sizes (Figure 4b), the ratio of surfactant-BSA to QDs surface can be regarded to be similar. Therefore, the corresponding physical quenching rate from MoS₂ QDs to ¹O₂ can be deemed to be the similar in this study.

The following discussion has been added in page 19.

“We note that the result of calculation was relatively external quantum yield. It is the combined result of several factors like the internal quantum efficiency and quenching effect from MoS₂. Several studies have shown that physical quenching exist between ¹O₂ and N-lone electron pair of amines, especially aromatic amine which is the component of amino acid in the surfactant-BSA⁵². Here the surfactant to QDs ratio was assumed to be same in three MoS₂ QDs (based on the size), the corresponding physical quenching rate from MoS₂ QDs to ¹O₂ was deemed to be the same.”

Other comments:

Page 18, line 379

"The quantum yield of 3O2 generated by photosensitizes..." I would guess that the authors mean "The quantum yield of 1O2 generated by photosensitizes..."

Responses: Thank you for spotting this typo mistake. We have corrected in the manuscript.

The authors seem to talk about ROS and 1O2 as if these were the same thing. ROS include 1O2, but also includes radicals generated by Type I processes. The authors only talk about 1O2 when they mean singlet oxygen and use ROS only when they mean singlet oxygen and free radical type processes.

Responses: Thank you for noticing this. We have amended the corresponding texts in the manuscript. When we wrote ROS, we are referring to the entire class. When we wrote 1O₂, we were referring only to singlet oxygen, which is the feature of Type II photodynamic process.

Page 5, line 106:

"Subsequent adjusting of pH by adding HCl activates the sulfur precursors..."

What was the pH adjusted to? Please be specific an give a value.

Responses: The pH value was adjusted till 6.

Page 5 lines 110-111:

"This ease paves the way for higher scalability..." Do the authors mean "This easily paves the way for higher scalability..."?

Response: Yes, thanks for the grammar correction. We changed "ease" to "easily" in the revised version.

Overall, this is a very nice paper that goes far beyond the typical phenomenological descriptions found in most papers on QD synthesis and properties. If the control experiments outlined above confirm the author's hypothesis that surface defects increase the $1O_2$ quantum yield of their QDs, this paper could be publishable in Nature Communications.

Response: Many thanks again for your nice comments. We are thankful for your kind suggestion of the additional experiments. Indeed, it has helped us improve on our findings.

The reviewer's comments are italicized.

Our responses to the reviewer's comments are in normal black fonts.

Our revisions in the manuscript are in green highlights.

Reviewer #2 (Remarks to the Author):

The manuscript "Library Construction of Defect Variable Bioactive Transition Metals Dichalcogenides Quantum Dots" describes a biomineralization assisted bottom-up strategy for the synthesis of a wide library of transition metal dichalcogenides (TMD) QDs. The reactions they used are very fast (10-20 s) with very soft conditions of room temperature, aqueous and atmosphere. Further study on MoS₂ QDs synthesized by the method for biomedical applications demonstrated that increased sulfur defects correlated well with increased oxidative stress generation from photodynamic effect in cancer cells. One of the most exciting advantages of the proposed method is the preparation of atomic surface defects of QDs, which allows researchers to investigate the nanoparticles defect effects on their functions when using in different field, because the atom defect of nanosurface is an open question in many fields. So, I recommend to publish this work at Nature Communication after minor revision.

The minor questions:

- 1. Could the absorbance band of MoS₂ be tuned to the optimized NIR light-transparent window of biomedical applications?*

Response: Thank you for your question. This is a good point for future work. We are thinking of future experiments where we can make hybrid TMD QDs of different transition metals on the same QDs with our synthesis method. We hypothesize that we might be able to shift the absorbance band to the NIR window.

- 2. Figure 6a, how long were HMVEC cells incubated with MoS₂ to evaluate the viability? What about the cell viability of SW480 after treatment of MoS₂?*

Response: Thank you for your question. The incubation time was 24 hours for HMVEC. The cell viability of SW480 with treatment of MoS₂ but without irradiation remained high (Figure 6c).

- 3. What about the cellular uptake and subcellular localization of MoS₂ in HMVEC and SW480?*

Response: Thank you for your question. We believe that there is real substantial uptake of the QDs because of its sheer small size and there was a clear PDT effect. Unfortunately we did not go so deep into the intracellular compartmentalization fate as the PDT part of this study was the proof of concept experiment. However, we will definitely want to go and understand the biological characteristics of these QDs in future studies.

- 3. Please comment of the details of the laser such as beam diameter, energy distribution in the beam, total power, etc. and how the irradiation experiments were performed (for instance, a region of xx cm² was irradiated). It is important for reproducibility.*

Response: Thank you for your reminder of the details of the laser. The diameter of the light beam is 1.2 cm with the power intensity of 100 mW/cm². For the measurement of ¹O₂ efficiency, 1 ml of the MoS₂ QDs solution with ABDA (10 μM) were transferred into a cuvette, which was

further exposed to the light irradiation perpendicularly (100 mW/cm^2). The absorption of the mixtures was recorded every 2 min. The light source irradiated a region of 1 cm^2 was irradiated.

4. *What's the average diameter of the 3D spheroids? Could the laser spot cover the spheroid?*

Response: Thank you for your question and point. The 3D spheroids started out as a consistent size of around $100 \mu\text{m}$ (Fig 6d). They are consistently sized right at the start because they are formed from an agarose non-attaching mould. With the light spot of around 1.2 cm in diameter, the light spot exceeds the entire size of spheroid.

5. *PDT is mainly applied to the superficial treatments, such as breast cancer and skin diseases. Please describe the experiment of MoS₂-based PDT in the in vivo model of SW480 colon cancer?*

Response: Thank you for your question. Actually, the PDT SW480 cell model is a proof of concept experiment to show the PDT potential of our QDs in a cell line. It does not really have any real bearing to clinical practice as yet or even in in vivo model of colon cancer; at least in this really preliminary stage of novel synthetic discovery of these defect variable QDs. However, if I can exact a highly speculative guess of its use in colon cancer treatment, one can mix the QDs in a gel and attach it to an endoscopy probe. The delivery and excitation can be done with the endoscope at the colon cancer area. But we hope that the field would be inspired to further advance these defect engineered TMDs QDs in more creative ways in nanomedicine applications.

The reviewer's comments are italicized.

Our responses to the reviewer's comments are in normal black fonts.

Our revisions in the manuscript are in green highlights.

Reviewer #3 (Remarks to the Author):

This manuscript describe the preparation of TMD QDs at room temperature in 10-20 s by the reaction of sodium chalcogenides with metal chlorides or oxides in BSA as surfactant. The authors also demonstrated the control over defects by using different ratios of chalcogen and metal. The defects are created by the inclusion of oxygen which replaces sulphur in the crystal lattice. Pristine and the defect samples were then investigated for anticancer oxidative stress generation. A strong correlation was shown between the degree of sulphur defects and photodynamic efficiency.

This work is highly significant as it provides a simple and quick method in aqueous solution for the preparation of TMD QDs which can be applied for the whole TMD family. The creation and the control over the defects and the relationship of the defects to their photodynamic efficiency is an important observation.

I am satisfied that all the results presented in this work are verified by the experimental evidence. The work is presented clearly with sufficient experimental detail and good discussion. I recommend its publication without any change.

Response: Many thanks for your very kind and supportive comments.

The reviewer's comments are italicized.

Our responses to the reviewer's comments are in normal black fonts.

Our revisions in the manuscript are in green highlights.

Reviewer #4 (Remarks to the Author):

This work reports on the bottom-up synthesis of transition metal dichalcogenide (TMDC) quantum dots (QDs), the role of defects in QDs on the production of reactive oxygen species, and the efficacy of these QDs to treat cancer cells. While the authors performed a large number of experiments in this work, the science would be better communicated if only the most important results were presented with greater detail in their analysis. In general, there are two narratives to this study: (i) the facile synthesis of TMDC QDs and (ii) the use of MoS₂ QDs for cancer treatment. While the facile synthesis of MoS₂ QDs (that are more monodisperse than those obtained through top-down methods) is an important development, questions (as outlined below) remain unanswered about the second part of this study. These questions lessen the potential impact of the manuscript.

Some specific suggestions/questions:

1. *A more expansive literature review would be helpful to understand the current status of bottom-up MoS₂ QD synthesis.*

Responses: Thank you for your suggestions. We have added more references in the manuscript to help with understanding the current status of bottom up MoS₂ QDs synthesis; namey Sensors and Actuators B 252 (2017) 183–190. Advanced Optical Materials 5.9 (2017): 1601021. ACS Nano 2018, 12, 751–758.

2. *Which phase are the MoS₂ QDs?*

Responses: The XRD patterns suggest the MoS₂ QDs were hexagonal 2H-MoS₂, which can also be indicated by the light yellow color (Generally speaking, metallic 1T-MoS₂ has a dark color).

3. *Regarding XPS quantification of chemical compositions, if the sulfurs on cysteine ligands are obscuring the Mo:S ratio, does the ratio change with the other ligands used in the study?*

Response: We have conducted the experiment of preparing MoS₂-DL QDs with Poly-Arg, which is thiol free. The XPS results confirmed that the quantification of Mo:S ratio increase to be 1:1.88.

4. *How would surface states contribute to the calculated band gap of 3.45 eV (as the authors mentioned around lines 144-146)?*

Response: The surface states was supposed to narrow the calculated optical band gap, which can also be observed in Figure S16. As the presence of localized defect states during the bottom-up protocol, this fact can be understood in terms of defect generated band-tailing effect (Physica B: Condensed Matter 240.1-2 (1997): 8-12; Science 332.6025 (2011): 77-81; ACS Energy Letters 2.11 (2017): 2616-2624.).

5. *The authors mention that they believe that one MoS₂ QD is interacting with one BSA molecule from DLS measurements. Have the authors tried to use NMR to better quantify the degree of ligand-binding to the surface of the QD?*

Response: Due to the complexity of amino acids in BSA, the NMR spectrum of the product is complicated. Instead, we have employed thermogravimetric analysis (TGA) to quantify the

degree of BSA binding on the surface of QD (Figure S5), which we believe is a more direct way of determining BSA binding on the surface of the QD.

Figure S5. TGA of BSA-coated MoS₂ QDs performed under inert nitrogen atmosphere.

Analyzing the result from TGA curves, three stages was observed: weight loss below 100 °C can be assigned to water; weight loss from 200 °C to 450 °C is mainly due to the thermal decomposition of BSA. The weight loss from 450 °C is correspond to the thermal decomposition of MoS₂ (*Composites Part A: Applied Science and Manufacturing* 94 (2017): 1-9.). The incompletely decomposed carbonaceous product under N₂ atmosphere at 450 °C was about 20 % of the total BSA amount according to the previous BSA report (*Journal of colloid and interface science* 389.1 (2013): 31-41). Then the total amount of BSA and MoS₂ was calculated to be around 7.1 mg and 2.4 mg, separately.

The mass of per 3.9 nm MoS₂ QDs (based on TEM results) and BSA molecule were further calculated:

$$m_{\text{MoS}_2} = \frac{4}{3} \pi r^3 \times \rho_{\text{MoS}_2} = \frac{4}{3} \pi (1.95 \text{ nm})^3 \times 5.06 \text{ g/cm}^3 = 1.5 \times 10^{-19} \text{ g MoS}_2/\text{QD}$$

$$m_{\text{BSA}} = \frac{MW_{\text{BSA}}}{6.022 \times 10^{23}} = 1.1 \times 10^{-19} \text{ g BSA/Molecule}$$

Then the ratio of N_{BSA} to N_{MoS_2} was calculated as:

$$\frac{N_{\text{BSA}}}{N_{\text{MoS}_2}} = \frac{7.1/m_{\text{BSA}}}{7.4/m_{\text{MoS}_2}} = 4.03$$

The ratio estimated from TGA result (4.03) was slightly greater than the DLS measurements but still at the same order of magnitude. Considering that our simple calculation above is based on a single size QDs average diameter, the actual QDs size distribution may also affect the number of BSA ligand binding on the surface.

6. Does the number of ligands present on the surface of the QDs change with defect density (or amount of sulfur)?

Response: The number of surfactants on the surface of particles were supposed to be mainly affected by particle size, which determine the surface coverage of surfactant on the nanocrystal surface (Nano Lett., DOI: 10.1021/acs.nanolett.8b02325). In the case of MoS₂, the degree of defect in MoS₂ QDs didn't significantly change the size distribution of particles (Figure 4b). The number of ligands on QDs surface was therefore unlikely to differ much.

7. *Does the photodynamic therapy selectively kill cancer cells? Did it kill the HMVEC cells as well?*

Response: Thank you for your good question. We however did not set out to do a selective killing study since the SW480 section of this work is merely a proof of concept that defect laden MoS₂ QDs are capable to killing and the killing is correlated to extent of defects. However, we believe that there is unlikely to have selectivity of MoS₂ QDs on its own. If it kills SW480 non selectively, then it is likely that HMVEC cells will be also killed with photodynamic therapy as collateral damage at the location of laser irradiation. However, for the most part of the blood circulatory journey if the QDs are intravenously introduced, it would have little overall detrimental effect on the endothelium (Fig 6a without irradiation) as there will not be any whole-body irradiation.

8. *Why were SW480 cells used as opposed to other cancer cell lines?*

Response: SW480 is a common cell line used in PDT studies where PDT agents are tested; both nanotechnology and small molecules PDT agents (Lasers Med Sci. 2018. doi: 10.1007/s10103-018-2524-7; Photodiagnosis Photodyn Ther. 2018, 23, 132-143; Nanomedicine (Lond). 2018, 605-624). Our observations of SW480 response to PDT is that it has a fairly linear response over a wide dynamic dose range. So it is neither overly sensitive nor overly resistant to PDT. Moreover, SW480 form spheroids easily and colon cancer is one of the most commonly occurring cancer (Small 2015, 11, 702-712)

9. *Have similar studies on cancer cells been done with other MoS₂ structures (nanodisks, nanosheets)?*

Response: Yes, recently nanomedicine field are beginning to explore the use of MoS₂ based nanomedicine (*Advanced Science* 4, 8 (2017): 1600540. *ACS Nano*, 2018, 2922–2938). But there have been no reports about defect engineering in MoS₂ structures for PDT purposes.

10. *The analysis of photoluminescence measurements (278-300) with regard to defects seems incomplete. The authors state that the trend in photoluminescence quantum yield is likely due to different surface defect states, and that the relatively higher quantum yield of MoS₂-DH QDs suggests they are the most defective. While there are instances where defects may enhance quantum yield, they are not necessarily directly related in this way, and are often times inversely related. The authors acknowledge that two possible emission peaks present in the spectrum may be attributed to “intrinsic state emission (electron-hole recombination) and defect state emission”, but no deconvolution of the peaks is done. By visual inspection, the PL from the intrinsic state emission is also significantly higher in the MoS₂-DH QDs (Figure 4d), the most defective and brightest QDs, and not just the possible defect emission peak.*

Response: Thank you for raising this very important point. The deconvolution of the peaks has been done (Figure 4d). We totally agree with you that the origin of the photoluminescence of many nanocrystals with regards to defect states is still unclear and the correlation (positive or negative) between photoluminescence and defects remains the subject of some scientific debates. For instance, in single-layer MoS₂, Saiful I. Khondaker etc. reported that by when increasing defect creation via oxygen plasma treatment, they found that PL gradually decrease to complete quenching. They explained their observation as a direct to indirect bandgap transition through oxygen plasma treatment in MoS₂ layers (*Journal of Physical Chemistry C* 118.36 (2014): 21258-21263). In another report, a contradictory observation where strong PL enhancement of MoS₂ could be achieved through defect engineering via mild oxygen plasma irradiation. In this controversial study, the authors proposed it as a conversion from trion to exciton based on oxygen adsorption (Nan, Haiyan, et al. "Strong photoluminescence enhancement of MoS₂ through defect engineering and oxygen bonding." *ACS nano* 8, 6 (2014):

5738-5745). However, both of these well cited studies are primarily reported on 2D MoS₂ nanosheets. In our case, we do not have any literature support for 0D MoS₂ QDs. You are also right. In order not to mislead our readers, in this revision, we reduced our claim to a mere observation and just tell the truth about its currently controversial status.

In the revised manuscript, we have changed to

“Interestingly, we observed a trend that with more defects, the photoluminescence quantum yield also increased.” (page 14)

and we have added more discussion about the intrinsic state emission and defect state emission. (page 15)

“The controversy of whether defect sites and photoluminescence quantum yields are positively or negatively correlated is still ongoing. The influences of surface defects on photoluminescence quantum yields in even smaller sized MoS₂ quantum dots is still largely unknown. Besides, from the photoluminescence spectrum, the intrinsic state emission of the highest to lowest defect sites groups, MoS₂-D_H to MoS₂-D_L QDs still play the leading role in the PL emission and show significantly higher efficiency in MoS₂-D_H QDs. One of the possible explanation is the passivation effect from oxygen atoms in MoS₂ crystalline. In the presence of oxygen, Bard found the PL of CdSe QDs could be enhanced by as much as a factor of 6, resulting from the surface passivation by oxygen on nanocrystalline surface (Myung, Noseung, Yoonjung Bae, and Allen J. Bard. "Enhancement of the photoluminescence of CdSe nanocrystals dispersed in CHCl₃ by oxygen passivation of surface states." *Nano Lett.* **3**, 747-749 (2003).). This phenomenon of oxygen atoms induced surface passivation of QDs has also been identified in many other semiconductor QDs (Jang, Eunjoon, et al. "Surface treatment to enhance the quantum efficiency of semiconductor nanocrystals." *J. Phys. Chem. B* **108**, 4597-4600 (2004). Jung, Dae-Ryong, et al. "Semiconductor nanoparticles with surface passivation and surface plasmon." *Electronic Materials Letters* **7**, 185 (2011)). The embedded oxygen atoms in the crystalline of MoS₂ structures, presumably played two roles; first by creating sulfur distortion defects which support defect state emission, second by forming Mo-S-O bond on the crystalline surface, which passivate it thus enabling the intrinsic state emission enhancement. Here preliminary experiment on the sulfur defect (in the form of Mo-O) reveal that the defect could increase PL intensity. However, substantial work such as ultrafast dynamics study remains to be done to explore these photophysics of MoS₂ QDs in greater detail.” (page 15)

11. The authors' text (Line 68) and Scheme 1 claim a universal bottom-up route to synthesis of TMD quantum dots. While seven compounds are prepared, this is not sufficient to claim a universal approach as indicated in the scheme.

Response: We have revised the tone to be possibly universal in the abstract. We have also modified the caption in Scheme 1 to illustrate that in reality, we have only shown a subset of all the possible combinations of transition metals with chalcogens.

12. Of the other TMD nanodots, the authors state "highly homogenous of size distribution with diameters below 10 nm" (Line 223), and histograms in Figure 3 are cut off at 7-10 nm. In the corresponding TEM images, there are particles of greater size than 10 nm based on the 100 nm scale bar. The authors note that some rod-like shape can be found in MoTe₂ samples, but if this is in reference to the larger particles, they are present in images of other TMD species as well. Are these particle aggregates? Additionally, in Figure 1b, it looks like the standard deviation of the QDs is around 25%. Does the Gaussian fit change with smaller bin sizes (0.25 nm as opposed to 1 nm)?

Response: Thank you for bringing up this good and fair point. *While the majority of the synthesized TMD particles size was within 10 nm, there are indeed some particles such as MoTe₂ rod with size above 10 nm.* Besides, some aggregation of the particles can be found from

TEM images. Here BSA was applied as the surfactant for proof of concept. With different TMD QDs showing different reaction activities which may need special chelating groups for size and morphology control, the corresponding surfactants for each TMD QDs still need to be carefully optimized. We have lessened our claim as *“highly homogeneous of size distribution with majority of the diameters to be below 10 nm”*.

The Gaussian distribution is based on the unbiased statistics from TEM images. While it is possible for us to quantitatively assigned smaller bin sizes of 0.25 nm, but the TEM images are not of high enough resolution to bin at 0.25nm resolution.

13. *In the discussion on the photosensitization by TMD NDs, beginning on Line 374, the authors argue, “sulfur defects in MoS₂ QDs tend to reduce the bandgap with lowering EST”. A reduced bandgap doesn’t necessarily lead to a smaller energetic splitting between the singlet and triplet states, and the splitting is only one factor determining intersystem crossing rate. Additionally, “singlet” and “triplet” states of semiconductor nanocrystals are not easily differentiable.*

Response: Thank you for point. We have change our statements and add more discussion in the manuscript. (page 21)

In our opinion, from the photophysics point of view, our synthesized defective MoS₂ QDs with excitation-dependent PL do bear some resemblance to carbon quantum dots (Pan, Dengyu, et al. "Hydrothermal route for cutting graphene sheets into blue - luminescent graphene quantum dots." *Advanced materials* 22.6 (2010): 734-738. Ge J, et al. A graphene quantum dot photodynamic therapy agent with high singlet oxygen generation. *Nat. Commun.* 5, 4596 (2014)). The material related property with the surface molecular states, which unfortunately cannot be treated as the typical semiconductor nanocrystals. Theory calculation predicted the existing of the bent of spin singlet and triplet states in MoS₂ monomer molecules (Spirko, Jeffery A., et al. "Electronic structure and reactivity of defect MoS₂: I. Relative stabilities of clusters and edges, and electronic surface states." *Surface science* 542.3 (2003): 192-204). In addition, in carbon dots, the excitation dependent PL features were thought to be the synergistic effects of quantum confinement, surface traps and edge states. Taking together the observations and analyses, to describe the defect induced enhancement of ¹O₂ generation in MoS₂ QDs, we proposed the *“singlet” and “triplet” states of MoS₂ QD structure models* following the typical photosensitization process, which was also adopted in previous electronic structure description of carbon dots.

We agree that “singlet” and “triplet” states in typical semiconductor nanocrystals are not easily differentiated. Previous work suggest that the triplet state is the lowest excited states of QDs (for example, in CdSe QDs). If we adhere to this view, the reduced bandgap narrows the excited states, which favor the extraction of the triplet excitons from photoexcited semiconductor QDs to ³O₂ via energy matching.

Besides a reduced bandgap, the spin-orbit perturbations (H_{SO}) in defective MoS₂ QDs could also affect the intersystem crossing. The vibronic coupling involved in Mo-S and Mo-O bonds would be significantly increased due to the increasing degree of defect. The large amount of atoms and decrease of size with increasing edge sites and dangling bonds in MoS₂ QDs would further enhance the likelihood of the vibrational modes thus promoting the intersystem crossing.

We also note that the real exciton splitting situation due to the presence of defects and quantum confinement effect of typical semiconductor nanocrystals behavior in MoS₂ QDs complicated their energy state which unfortunately could not be addressed decisively in this manuscript. Further investigations to understand the electronic structure and exciton of MoS₂ QDs are being carried out.

14. *Line 187: The authors state “FWHM value” when giving the mean particle size.*

Response: Thank you for spotting this mistake. We have changed it to size.

Reviewers' comments:

Reviewer #1 (Remarks to the Author):

This is a revised version of a previously submitted manuscript; my comments are primarily dealing with the revisions I had previously suggested.

Unfortunately the manuscript is still not publishable, as the authors failed to do several simple controls that I had suggested. In fact, the only control that was (satisfactorily) done was to check the photosensitivity of the singlet oxygen trap (by ^1H NMR and HPLC) Furthermore, while I understand that revisions are often done under significant time pressure, the writing and especially the imprecise and incorrect use of technical terminology in the revised sections requires significant revision.

Specifically, I had suggested that the authors follow the disappearance of the singlet oxygen trap by ^1H NMR (in the presence of the QDs) to verify that the endoperoxide of the trap is formed. Right now, we only have UV/vis data, which shows the disappearance of the trap, but we really need to determine that an endoperoxide is formed to verify that singlet oxygen was the reactive species involved in the disappearance of the trap. Of course it would also be fine if the authors can come up with another method that conclusively shows formation of an endoperoxide (and not some other species) from irradiation of the QDs in the presence of O_2 .

The authors state that "Combined the experiment and analysis above we believe the involved photooxidation was mainly due to Type II process", but in fact no experiments were done to verify that this is a type II process, and the anthracene derivative could also be oxidized in a radical process. (QDs are well known to initiate both type I and type II processes. radical could also be formed from the amino groups on the surface, see below) I had suggested the simple standard test for a singlet oxygen process vs. type I, namely running the trapping experiment under identical conditions in deuterated and non-deuterated solvents. A significant increase of photooxidation of the trap in deuterated solvents would be indicative of a singlet O_2 process, due to the longer lifetime of $^1\text{O}_2$ in deuterated solvents. This is all standard methodology in the field of singlet oxygen chemistry and frankly not very difficult to do, so I am puzzled as to why this experiment was not performed.

Related to the previous paragraph: The authors now state "As an anthracene derivative, ABDA has been widely used as $^1\text{O}_2$ -specific probe

acting as both a physical and chemical quencher for 1O_2 , which are typically generated in type II photodynamic process^{46, 47, 48}." (lines 352-354). ABDA is indicative of singlet O_2 formation if an endoperoxide product is formed. We have no data currently as to whether or not that was the case. I suspect that this species is in fact formed, but we cannot do science based on guesses - we must perform appropriate tests and controls. Furthermore, is ABDA in fact a physical quencher of singlet O_2 as the authors claim in the quoted paragraph? If so what is the quenching rate constant (k_q) for physical quenching, especially relative to chemical reaction (k_r)? If ABDA were a significant physical quencher of 1O_2 , then it would not be able to trap a significant fraction of 1O_2 (physical quenching = conversion to ground state O_2 without oxidation of the quencher).

Also: What do the authors mean by employing the plural in the above quote ("...which are typically generated...")? Does this refer to singlet O_2 (which should be singular)? Or does this refer to different quenching processes (in which case the quoted statement makes no sense). And what is meant by "typically"? Are there any type II processes that do not involve singlet O_2 ?

The new section below (lines 380 -387) also has numerous problems:

"The MoS₂-DH and MoS₂-DM groups respectively exhibited 1O_2 quantum yield of about 2.29 times and 1.64 times of MoS₂-DL QDs' quantum yield (Figure 5c). The calculated values are external quantum yields consisting of internal quantum efficiency and quenching effect from MoS₂. Moreover, this physical quenching exists between 1O_2 and N-lone pair electron of amines, especially aromatic amines which is a major component in the BSA-surfactant which appears to be of equal amounts on the QDs groups (based on the size)⁵². Thus suggesting that the physical quenching can be deemed to be the same in three QDs and the increasing defects does increase quantum yield of 1O_2 ."

First of all, given the large amount physical quenching by amines (and the fact that the QDs are not completely homogeneous), there is no way that the quantum yield differences can be stated to three significant figures (2.29 vs 1.64)! Good scientific practice would be to do multiple measurements, and report the error as one standard deviation (and use appropriate significant figures based on this error). It pains me to have to point this out in a review of a Nature paper, although sadly this kind of sloppiness is very common nowadays and certainly not limited to the authors of this work.

Also: What are "external quantum yields"? How is that defined as opposed to the standard quantum yield of singlet O_2 generation (which is the product of triplet formation times triplet quenching by 3O_2 times the fraction of the latter process that leads to 1O_2 formation)? And what is meant by the "internal quantum efficiency" - of what? Singlet oxygen formation? Triplet QD formation? The authors need to understand and properly define the terminology that they use.

Furthermore: The last statement of the above paragraph (starting with "Thus") seems to be missing half a sentence. The authors claim that physical quenching can be "deemed" to be the same - but this would only be true if the same number of unprotonated amino groups were on each QD. Is that

the case? If so, it needs to be stated. If not, comparison of the quantum yield data would be pointless, given the very large rate constants for physical quenching by aromatic amines (at least $10^8 \text{ M}^{-1}\text{sec}^{-1}$) Perhaps the authors can estimate (not deem) to what extent the number of amino groups are similar. The entire paragraph needs to be rewritten, with special attention to using correct technical terminology.

Overall, I think the paper can probably still be published at a later stage, but the problems outlined above do need to be addressed.

Reviewer #2 (Remarks to the Author):

The authorship have made proper revisions.

Reviewer #4 (Remarks to the Author):

The authors have satisfactorily addressed my concerns over the scope of several claims made during discussion and analysis. They have also conducted control experiments regarding illumination of ABDA alone, correctly recommended by Reviewer 1, but I note that the authors do not directly probe product formation by ^1H NMR as Reviewer 1 requested. Overall, I believe the work is publishable, but my main concern with regards to publication in Nature Communications is the novelty and impact of the work, which I believe is diminished due to similar studies using other MoS₂ nanostructures as cited in their answer to Question 9 of Reviewer 4 (Advanced Science 4, 8 (2017): 1600540. ACS Nano, 2018, 2922–2938).

The reviewer's comments are italicized.

Our responses to the reviewer's comments are in normal black fonts.

Our revisions in the manuscript are in green highlights.

Reviewer #1 (Remarks to the Author):

This is a revised version of a previously submitted manuscript; my comments are primarily dealing with the revisions I had previously suggested.

Unfortunately the manuscript is still not publishable, as the authors failed to do several simple controls that I had suggested. In fact, the only control that was (satisfactorily) done was to check the photosensitivity of the singlet oxygen trap (by ^1H NMR and HPLC) Furthermore, while I understand that revisions are often done under significant time pressure, the writing and especially the imprecise and incorrect use of technical terminology in the revised sections requires significant revision.

Specifically, I had suggested that the authors follow the disappearance of the singlet oxygen trap by ^1H NMR (in the presence of the QDs) to verify that the endoperoxide of the trap is formed. Right now, we only have UV/vis data, which shows the disappearance of the trap, but we really need to determine that an endoperoxide is formed to verify that singlet oxygen was the reactive species involved in the disappearance of the trap. Of course it would also be fine if the authors can come up with another method that conclusively shows formation of an endoperoxide (and not some other species) from irradiation of the QDs in the presence of O_2 .

Response: We are sorry that we did not complete each and every control that was mentioned in the first revision. In this revision, we have done all the suggested experiments and many thanks for giving us opportunities to improve on our manuscript and story.

We heeded your suggestions. As a control, we used Rose Bengal (RB). RB is known to generate $^1\text{O}_2$ under irradiation. When mixed with ABDA and irradiated, ABDA reacts to form ABDA endoperoxide. That change from unreacted ABDA itself to ABDA endoperoxide can be detected with ^1H NMR. The right shift of the spectra is most likely due to the photochemical reaction on the central aromatic ring of anthracene of ABDA (to ABDA endoperoxide). We then used this RB photochemical reaction with ABDA as a reference control. We determined the shift in the ABDA spectra for RB (irradiated vs non-irradiated) and for MoS_2 QDs (irradiated vs non-irradiated). We found that the right shift arising from irradiation of either RB or MoS_2 QDs, (the H peaks at $\delta = 8.26, 7.52$ and 4.06 ppm were right shifted to $\delta = 7.40, 7.30$ and 3.48 ppm) were similar in the reaction between RB with ABDA and MoS_2 with ABDA (new Figure S16a-b with S16 c-d). Since it is the similar ABDA endoperoxide that was produced, then the reactive species known to be formed after RB irradiation should be the same as that of MoS_2 QDs irradiation.

Figure S16. $^1\text{H-NMR}$ spectrum of the product of ABDA with and without light irradiation: (a) Incubated with RB without irradiation, (b) Incubated with RB with irradiation, (c) Incubated with MoS_2 QDs without irradiation and (d) Incubated with MoS_2 QDs with irradiation. For the MoS_2 QDs incubation group, the MoS_2 QDs were removed by using centrifugal filters (MWCO 10K) before $^1\text{H-NMR}$ testing.

The authors state that "Combined the experiment and analysis above we believe the involved photooxidation was mainly due to Type II process", but in fact no experiments were done to verify that this is a type II process, and the anthracene derivative could also be oxidized in a radical process. (QDs are well known to initiate both type I and type II processes. radical could also be formed from the amino groups on the surface, see below) I had suggested the simple standard test for a singlet oxygen process vs. type I, namely running the trapping experiment under identical conditions in deuterated and non-deuterated solvents. A significant increase of photooxidation of the trap in deuterated solvents would be indicative of a singlet O_2 process, due to the longer lifetime of $^1\text{O}_2$ in deuterated solvents. This is all standard methodology in the field of singlet oxygen chemistry and frankly not very difficult to do, so I am puzzled as to why this experiment was not performed.

Response: Yes, it is a straightforward experiment to do. We managed to overcome some issue with the getting the deuterated water this time round. So for this revision, we did the suggested experiment.

For the ABDA in deuterated vs in non-deuterated experiment, the photooxidation of the trap was carried out in pure H_2O and D_2O separately. Compared to the reaction carried out in non-deuterated water, a significant decrease of absorbance at 400 nm in D_2O was observed which is in line with what you have

kindly pointed out. (**Figure S17**), ie any formed $^1\text{O}_2$ has a longer lifetime in D_2O so will react with more ABDA.

On top of the above suggested experiment, we added on one more unsolicited experiments to confirm our thoughts on singlet $^1\text{O}_2$. We used electron spin resonance spectrum (ESR) to examine the photosensitization process by employing 2, 2, 6, 6-tetramethylpiperidine (TEMP) as $^1\text{O}_2$ trapping agent. As shown in **Figure S18**, MoS_2 was first mixed with TEMP without irradiation and the ESR spectra was recorded. After irradiation and 5 mins of reaction time, the TEMP's ESR was again measured and was found to be showing significantly higher 1:1:1 triplet signal characteristic with g-value of 2.005, which is consistent with the characteristic of 2, 2, 6, 6-tetramethylpiperidine-N-oxyl (TEMPO) ESR signal. At the same time, the untreated control group of just TEMP at the same concentration showed no increase under the identical irradiation condition. These results collectively suggest that the irradiation of MoS_2 QDs can induce the formation of $^1\text{O}_2$. So we believe that our QDs at least produces Type II singlet $^1\text{O}_2$ oxidative stress but we have not excluded or included Type I process.

Figure S17. Absorption spectrum of the mixture solution of $\text{MoS}_2\text{-D}_L$ QDs and ABDA in dH_2O (black curve) and D_2O (blue curve) before and after 8 min of light exposure.

Figure S18. ESR spectrum of MoS_2 QDs in the presence of TEMP.

We revised the write up in the main text as follows:

“The product of the ABDA trap after MoS_2 QDs irradiation was checked by ^1H -NMR and compared with the corresponding product of Rose Bengal (RB), a positive control known to generate $^1\text{O}_2$ under irradiation. Similar chemical shifts were also observed in the corresponding H peaks of the products after irradiation (**Figure S16**), suggesting that the species generated from MoS_2 QDs and RB irradiation reacted with ABDA and produced similar ABDA products. We checked again with repeating the irradiation of $\text{MoS}_2\text{-ABDA}$ reaction in D_2O or H_2O conditions. It was found that there was higher depletion of the ABDA substrate when using D_2O while irradiating MoS_2 QDs (**Figure S17**). This showed that $^1\text{O}_2$ was generated from

irradiation of MoS₂ QDs. We further checked with a ¹O₂ 2, 2, 6, 6-tetramethylpiperidine (TEMP) sensor-electron spin resonance spectrum (ESR) assay. We found more product after 5 minutes of MoS₂ irradiation (**Figure S18**). Collectively, ¹O₂ was likely generated after MoS₂ irradiation."

Related to the previous paragraph: The authors now state "As an anthracene derivative, ABDA has been widely used as ¹O₂-specific probe acting as both a physical and chemical quencher for ¹O₂, which are typically generated in type II photodynamic process^{46, 47, 48.}" (lines 352-354). ABDA is indicative of singlet O₂ formation if an endoperoxide product is formed. We have no data currently as to whether or not that was the case. I suspect that this species is in fact formed, but we cannot do science based on guesses - we must perform appropriate tests and controls.

Response: We apologize for the previous revision. Now the endoperoxide product was experimentally verified by NMR and compared with the product from RB under irradiation (**Figure S16**). The trapping experiments were also conducted and compared in pure water and D₂O separately (Figure S17) and the additional ESR TEMP experiment (Figure S18) indicated the ¹O₂ was indeed formed.

Furthermore, is ABDA in fact a physical quencher of singlet O₂ as the authors claim in the quoted paragraph? If so what is the quenching rate constant (k_q) for physical quenching, especially relative to chemical reaction (k_r)? If ABDA were a significant physical quencher of ¹O₂, then it would not be able to trap a significant fraction of ¹O₂ (physical quenching = conversion to ground state O₂ without oxidation of the quencher).

Response: Your insights help this manuscript. While two papers described the photolytic transformations of ABDA with a generally used scheme for the sensitized photooxidation with singlet oxygen, namely chemical reaction, physical quenching and solvent induced deactivation, they didn't show the direct evidence that ABDA is in fact a physical quencher and the ratio of k_q/k_r (*Russ. J. Gen. Chem.* **71**, 36-41 (2001). *Photochem. Photobiol.* **90**, 85-91(2014)). We apologize for making that statement as we are not experts in photochemistry but since this manuscript is not really about ABDA, we have deleted those statements. Many thanks for highlight this error for us.

Also: What do the authors mean by employing the plural in the above quote ("...which are typically generated...")? Does this refer to singlet O₂ (which should be singular)? Or does this refer to different quenching processes (in which case the quoted statement makes no sense). And what is meant by "typically"? Are there any type II processes that do not involve singlet O₂?

Response: From your detailed scrutiny, we realized that for this part we could not be exactly clear on the behavior of ABDA from literature. So we have deleted that statement on ABDA: ". As an anthracene derivative, ABDA has been widely used as ¹O₂-specific probe acting as both a physical and chemical quencher for ¹O₂, which are typically generated in type II photodynamic process^{46, 47, 48.}"

*The new section below (lines 380 -387) also has numerous problems:
"The MoS₂-DH and MoS₂-DM groups respectively exhibited ¹O₂ quantum yield of about 2.29 times and 1.64 times of MoS₂-DL QDs' quantum yield (Figure 5c). The calculated values are external quantum yields consisting of internal quantum efficiency and quenching effect from MoS₂. Moreover, this physical quenching exists between ¹O₂ and N-lone pair electron of amines, especially aromatic amines which is a major component in the BSA-surfactant which appears to be of equal amounts on the QDs groups (based on the size)⁵². Thus suggesting that the physical quenching can be deemed to the same in three QDs and the increasing defects does increase quantum yield of ¹O₂."*

First of all, given the large amount physical quenching by amines (and the fact that the QDs are not completely homogeneous), there is no way that the quantum yield differences can be stated to three significant figures (2.29 vs 1.64)! Good scientific practice would be to do multiple measurements, and report the error as one standard deviation (and use appropriate significant figures based on this error). It pains me to have to point this out in a review of a Nature paper, although sadly this kind of sloppiness is very common nowadays and certainly not limited to the authors of this work.

Response: Thank you for pointing out this. We pay attention to this. More measurements were carried out and applied for the calculation. The results were plotted as standard deviation now.

Figure 5(c) Relative $^1\text{O}_2$ quantum yield of two MoS₂ QDs groups relative to MoS₂-D_L QDs. Quantitative data are mean \pm SD, n=4, Student's t-test, $P^* < 0.05$.

The corresponding kinetic curves of three MoS₂ QDs with separate experiments were put in the SI.

Figure S19. Decomposition rate of the photosensitizing process of MoS₂-D_H (a), MoS₂-D_M (b) and MoS₂-D_L (c) samples experimental time course runs (n=4), where A₀ is the absorbance of initial absorbance of ABDA and A is the absorbance of ABDA under light irradiation at different time points.

Also: What are "external quantum yields"? How is that defined as opposed to the standard quantum yield of singlet O₂ generation (which is the product of triplet formation times triplet quenching by $^3\text{O}_2$ times the fraction of the latter process that leads to $^1\text{O}_2$ formation)? And what is meant by the "internal quantum efficiency" - of what? Singlet oxygen formation? Triplet QD formation? The authors need to understand and properly define the terminology that they use.

Furthermore: The last statement of the above paragraph (starting with "Thus") seems to be missing half a sentence. The authors claim that physical quenching can be "deemed" to be the same - but this would only be true if the same number of unprotonated amino groups were on each QD. Is that the case? If so, it needs to be stated. If not, comparison of the quantum yield data would be pointless, given the very large rate constants for physical quenching by aromatic amines (at least $10^8 \text{ M}^{-1}\text{sec}^{-1}$). Perhaps the authors can estimate (not deem) to what extent the number of amino groups are similar. The entire paragraph needs to be rewritten, with special attention to using correct technical terminology.

Response: Thank you for your explanation for the differences in the terminology. We have learnt a lot about oxidative chemistry from your comments. We have made the corresponding corrections in the manuscript with the hope that we got it right now.

With regards to your questions on the internal quantum efficiency and external quantum yield, we are sorry that we have misunderstood and used those terms from solar cells field to describe here. We acknowledge our error and removed it from the manuscript. But your points of BSA is well taken and we did some experiments to verify the similarity in the amount of amine groups on the QDs. For the estimation of amino groups, as the amino groups are derived from BSA in the synthesis process, here we quantified the amount of BSA on three kinds of QDs to compare the difference of amino groups on MoS₂ dots. The three kinds of MoS₂ QDs were quantified by ICP-OES and adjusted to the same concentration. Then the amount of BSA on MoS₂ dots was tested with Pierce™ BCA protein assay kit. The results show that there is no significant difference of BSA on three MoS₂ QDs, suggestive that number of amino groups found on the three kinds of MoS₂ QDs were similar.

The “. Thus” should have been “, thus”. We have made that change in the main text. Thank you for your careful reading.

Figure S20. (a) The quantification curve for BSA using the standard test tube protocol (37 °C/30 min incubation); (b) There is minimal difference between the BSA amounts on three MoS₂ QDs.

The entire paragraph has been rewritten as:

The MoS₂-D_H and MoS₂-D_M groups respectively exhibited ¹O₂ quantum yield of approximately 2.3 times and 1.7 times of MoS₂-D_L QDs' quantum yield (**Figure 5c**, **Figure S19**). Physical quenching between ¹O₂ and N's lone pair electrons of amines may exist; especially those aromatic amines of the BSA-surfactant⁴³. It is therefore important to check that the differences in ¹O₂ quantum yield is not due to different amounts of BSA that are on the surface of the three kinds of MoS₂ QDs. We quantified the amount of BSA on the surface of the three kinds of MoS₂ using the micro-BCA protein assay. The BCA protein based assay showed no significantly different amounts of BSA on the same amount of the three MoS₂ QDs (**Figure S20**). This indicated that the physical quenching due to proteins on the surface of the three MoS₂ QDs groups are similar. Thus, confirming that the significant increase in the ¹O₂ quantum yields of MoS₂-D_M and MoS₂-D_H over MoS₂-D_L is due to increasing defects.

Overall, I think the paper can probably still be published at a later stage, but the problems outlined above do need to be addressed.

Response: Thank you for your time, guidance and patience. We have learnt a lot.

Reviewer #4 (Remarks to the Author):

The authors have satisfactorily addressed my concerns over the scope of several claims made during discussion and analysis. They have also conducted control experiments regarding illumination of ABDA alone, correctly recommended by Reviewer 1, but I note that the authors do not directly probe product formation by ^1H NMR as Reviewer 1 requested. Overall, I believe the work is publishable, but my main concern with regards to publication in Nature Communications is the novelty and impact of the work, which I believe is diminished due to similar studies using other MoS₂ nanostructures as cited in their answer to Question 9 of Reviewer 4 (Advanced Science 4, 8 (2017): 1600540. ACS Nano, 2018, 2922–2938).

Response: Thank you for your time. We are greatly to know that our discussion and analysis satisfied you. The endoperoxide products have been checked by ^1H NMR now (Figure S16) and more (Figure S17 and S18). At the same time we can't agree with that our work is similar to the two studies in our previous response and therefore have compromised with our novelty. While the two works were mainly about the usage of already established protocols of 2D-MoS₂ nanosheets (top down exfoliation approach - ACS Nano pp) or 80nm big MoS₂ nanoflowers (Adv Sci pp) preparation for bioapplications, our work develops a biomineralization assisted bottom-up strategy that can be broadly applied for the synthesis of a wide library of TMD quantum dots and further exploited the photodynamic efficiency with MoS₂ QDs as an example of defect engineering. Moreover, since the ACS Nano is based on top down exfoliation methods to obtain the 2D-MoS₂ nanosheets, there is no attempt to defect engineering. The same goes for the Adv Sci paper. Both papers center on the applications while we have centered on the science of the first aqueous based facile, mild conditions bottom up synthesis with defect engineering capabilities. So those two very nice papers are a different group of work from our work and we are very confident of this paper's novelty.

In short, the novelty of our work is the following:

1. This is the first work using a facile (~10-20 seconds), aqueous and atmospheric conditions to establish a library of various transition metals with dichalcogenides at room temperature via bottom-up route.
2. Our bottom-up approach enables the modulation of actual synthesized stoichiometry to be different from the molecular stoichiometry, allowing fast defect engineering in transition metals dichalcogenides QDs.
3. Furthermore, we reveal the potential reaction pathway of MoS₂ QDs and the defective crystal structures, which might be $(\text{MoO}_4)^{2-} + \text{S}^{2-} \rightarrow \text{MoO}_x\text{S}_{2-x}$.
4. With MoS₂ QDs as an example, this is the first experimental demonstration of the correlation between degree of defects on QDs and photodynamic efficiencies.

But we have cited the stated two references as ref 3 and 4 as more papers where TMDs nanomaterials are emerging applications in the biology and biomedical domains.

REVIEWERS' COMMENTS:

Reviewer #1 (Remarks to the Author):

This is a much improved version of the previously reviewed manuscript, and I am pleased to recommend publication of this work. The authors completed all necessary control experiments and did a nice job rewriting parts of the manuscript as suggested.

I would also like to state that I am very impressed by the number of experiments completed by the co-author(s) of this paper in a very short time-frame for the review process. The paper will be a valuable contribution to the literature on synthesis and photophysical properties of quantum dots.